The EMBO Journal (2013) 32, 2708–2721
www.embojournal.org

# Unlimited *in vitro* expansion of adult bi-potent pancreas progenitors through the Lgr5/R-spondin axis

**Meritxell Huch**[1,4], **Paola Bonfanti**[2,4], **Sylvia F Boj**[1,4], **Toshiro Sato**[1,4,5], **Cindy JM Loomans**[1,3], **Marc van de Wetering**[1], **Mozhdeh Sojoodi**[2], **Vivian SW Li**[1,6], **Jurian Schuijers**[1], **Ana Gracanin**[1], **Femke Ringnalda**[1,3], **Harry Begthel**[1], **Karien Hamer**[1], **Joyce Mulder**[1], **Johan H van Es**[1], **Eelco de Koning**[1,3], **Robert GJ Vries**[1], **Harry Heimberg**[2,4,*] **and Hans Clevers**[1,4,*]

[1]Hubrecht Institute for Developmental Biology and Stem Cell Research, University Medical Centre Utrecht, Utrecht, The Netherlands, [2]Diabetes Research Center, Vrije Universiteit Brussel, Brussels, Belgium and [3]Department of Nephrology, Leiden University Medical Center, Albinusdreef 2, Leiden, The Netherlands

*Lgr5* marks adult stem cells in multiple adult organs and is a receptor for the Wnt-agonistic R-spondins (RSPOs). Intestinal, stomach and liver *Lgr5*[+] stem cells grow in 3D cultures to form ever-expanding organoids, which resemble the tissues of origin. Wnt signalling is inactive and *Lgr5* is not expressed under physiological conditions in the adult pancreas. However, we now report that the Wnt pathway is robustly activated upon injury by partial duct ligation (PDL), concomitant with the appearance of *Lgr5* expression in regenerating pancreatic ducts. *In vitro*, duct fragments from mouse pancreas initiate *Lgr5* expression in RSPO1-based cultures, and develop into budding cyst-like structures (organoids) that expand five-fold weekly for >40 weeks. Single isolated duct cells can also be cultured into pancreatic organoids, containing *Lgr5* stem/progenitor cells that can be clonally expanded. Clonal pancreas organoids can be induced to differentiate into duct as well as endocrine cells upon transplantation, thus proving their bi-potentiality.

*The EMBO Journal* (2013) **32**, 2708–2721. doi:10.1038/emboj.2013.204; Published online 17 September 2013
*Subject Categories:* development; molecular biology of disease

*Corresponding authors. H Clevers, Hubrecht Institute for Developmental Biology and Stem Cell Research, Uppsalalaan 8, 3584CT Utrecht & University Medical Centre Utrecht, Utrecht, The Netherlands. Tel.: +31 30 212 1800; Fax: +31 30 251 6464; E-mail: h.clevers@hubrecht.eu or H Heimberg, Diabetes Research Center, Vrije Universiteit Brussel, Laarbeeklaan 103, D2, B1090 Brussel, Belgium. Tel.: +3224774477; Fax: +3224774472; E-mail: Harry.Heimberg@vub.ac.be
[4]These authors contributed equally to this work.
[5]Present address: Department of Gastroenterology, School of Medicine, Keio University, 35 Shinanomachi, Shinnjukuku, Tokyo 160-8582, Japan
[6]Present address: Division of Stem Cell Biology and Developmental Genetics, MRC National Institute for Medical Research, The Ridgeway, Mill Hill, London NW7 1AA, UK

*Keywords*: beta cell; duct cell; pancreas; Wnt; stem cell

## Introduction

As first demonstrated for intestinal crypts (Korinek *et al*, 1998), Wnt signalling plays a crucial role in the regulation of multiple types of adult stem cells and progenitors (Clevers and Nusse, 2012). The Wnt target gene *Lgr5* marks actively dividing stem cells in Wnt-driven, continuously self-renewing tissues such as small intestine and colon (Barker *et al*, 2007), stomach (Barker *et al*, 2010) and hair follicles (Jaks *et al*, 2008). However, expression of *Lgr5* is not observed in endodermal organs with a low rate of spontaneous self-renewal, such as liver or pancreas. In the liver, we have recently described that Wnt signalling is highly activated during the regenerative response following liver damage. *Lgr5* marks an injury-induced population of liver progenitor cells capable of regenerating the tissue after injury (Huch *et al*, 2013).

In the adult pancreas, Wnt signalling is inactive (Pasca di Magliano *et al*, 2007), yet it is essential for its development during embryogenesis (Murtaugh *et al*, 2005; Heiser *et al*, 2006). The embryonic pancreas harbours multipotent progenitor cells that can give rise to all pancreatic lineages (acinar, duct and endocrine) (Zaret and Grompe, 2008). Injury to the pancreas can reactivate the formation of new pancreatic islets, called islet neogenesis, by mechanisms still not entirely understood but that resemble development of the embryonic pancreas (Bouwens, 1998; Gu *et al*, 2003). Lineage tracing studies have demonstrated that these 'de novo beta cells' can be derived from pre-existing beta cells (Dor *et al*, 2004), or by conversion of alpha cells, after almost 90% beta-cell ablation (Thorel *et al*, 2010). Also, severe damage to the pancreas, by means of partial duct ligation (PDL) or acinar ablation, can stimulate non-endocrine precursors, such as duct cells, to proliferate and differentiate towards acinar (Criscimanna *et al*, 2011; Furuyama *et al*, 2011), duct (Criscimanna *et al*, 2011; Furuyama *et al*, 2011; Kopp *et al*, 2011) and also endocrine lineages (including beta cells) (Xu *et al*, 2008; Criscimanna *et al*, 2011; Pan *et al*, 2013; Van de Casteele *et al*, 2013), suggesting the existence of a pancreas progenitor pool within the ductal tree of the adult pancreas.

The development of a primary culture system based on the adult, non-transformed progenitor pancreas cells would represent an essential step in the study of the relationships between pancreas progenitor cells, their descendants and the signals required to instruct them into a particular lineage fate. Also, the production of an unlimited supply of adult pancreas cells would facilitate the development of efficient cell replacement therapies. Most of the available pancreas adult stem cell-based culture protocols yield cell populations that undergo senescence over time unless the cells become transformed.

It is fair to say that no robust, long-term culture system exists today that is capable of maintaining potent, clonal expansion of adult non-transformed pancreas progenitors over long periods of time under defined conditions. Recently, endoderm progenitors derived from embryonic stem cells (ESCs) (Cheng *et al*, 2012; Sneddon *et al*, 2012) or induced pluriportent stem cells (iPSCs) (Cheng *et al*, 2012) were serially expanded, in co-culture with pancreas mesenchyme or MEFs, respectively, and gave rise to glucose-responsive beta cells *in vitro* (Cheng *et al*, 2012) and glucose-sensing and insulin-secreting cells, when transplanted, *in vivo* (Sneddon *et al*, 2012).

We have recently described a 3D culture system that allows long-term expansion of adult small intestine, stomach and liver cells without the need of a mesenchymal niche, while preserving the characteristics of the original adult epithelium (Sato *et al*, 2009; Barker *et al*, 2010; Huch *et al*, 2013). A crucial component of this culture medium is the Wnt agonist RSPO1

(Kim *et al*, 2005; Blaydon *et al*, 2006), the recently reported ligand of *Lgr5* and its homologues (Carmon *et al*, 2011; de Lau *et al*, 2011). Here, we describe that Wnt signalling and *Lgr5* are strongly upregulated in remodelling duct-like structures upon injury by PDL. We exploit the Wnt-Lgr5-Rspo signalling axis to generate culture conditions that allow long-term expansion of adult pancreatic duct cells, which maintain the ability to differentiate towards both duct and endocrine lineages when provided the proper signals.

## Results

### Wnt signalling and Lgr5 expression are upregulated during pancreas regeneration following PDL

We first sought to document Wnt pathway activation in normal adult pancreas and following acute damage. We used the *Axin2-LacZ* allele as a general reporter for Wnt

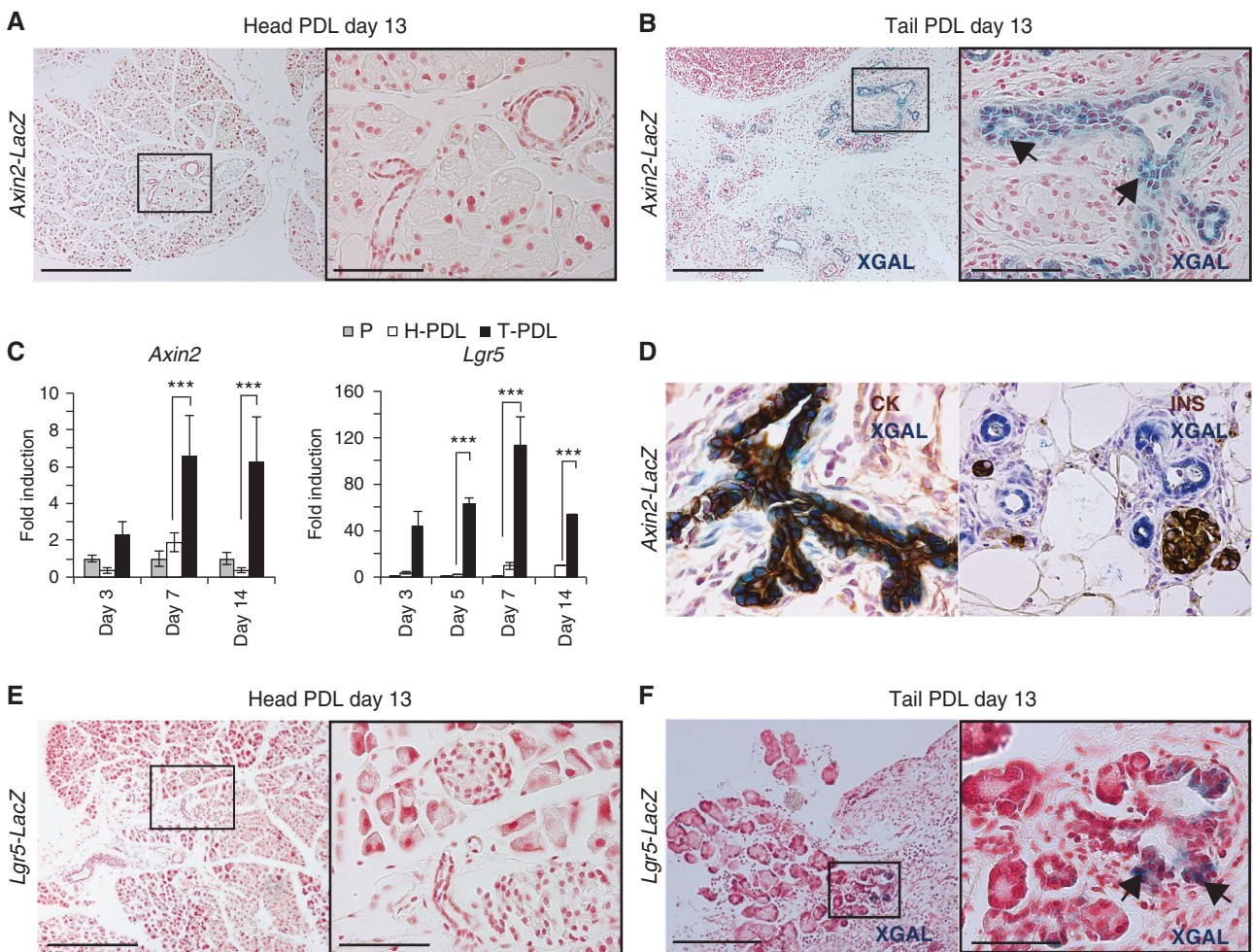

**Figure 1** Induction of *Axin2* and *Lgr5* expression upon damage on adult pancreas. (**A**, **B**) *Axin2-LacZ* induction in newly formed pancreatic ducts upon PDL. *Axin2-LacZ* mice ($n = 6$) underwent PDL as explained in Materials and methods. Mice were sacrificed at the indicated time points and the non-ligated pancreatic tissue (Head-PDL) was separated from the ligated part (Tail-PDL). (**A**) Head-PDL and (**B**) Tail-PDL portion 13 days after injury. Arrows indicate XGAL-specific staining exclusively detected in the pancreatic ducts of the ligated pancreas. Scale bars 200 µm (**A**, **B**, left panels) and 50 µm (**A**, **B**, right panels). (**C**) qPCR analysis of *Axin2* and *Lgr5* mRNA in adult pancreas following PDL. Results are represented as mean ± s.e.m. of at least three independent experiments. The *Hprt* housekeeping gene was used to normalize for differences in RNA input. Non-parametric Mann–Whitney test was used. ***$P < 0.0001$. P, pancreas from a sham-operated mice; H-PDL, Head-PDL (non-affected area after PDL injury); T-PDL, Tail-PDL (affected area after PDL injury). (**D**) Representative image of a XGAL staining on an *Axin2-LacZ* pancreas after PDL, sections were co-stained either for pancytokeratin (CK), a duct cell marker, or for insulin (INS), an endocrine β-cell marker. XGAL staining (reflecting *Axin2* expression) was detected exclusively in the pancreatic duct compartment. (**E**, **F**) *Lgr5-LacZ* induction in the ductal tree upon PDL. (**E**) Head PDL pancreas ($n = 6$) do not show XGAL staining, indicating that *Lgr5* is not expressed in non-injured pancreas. (**F**) *Lgr5* reporter is detected (arrows) in the Tail-PDL portion of the ligated pancreas ($n = 6$). Scale bars 200 µm (**E**, **F**, left panels) and 30 µm (**E**, **F**, right panels).

signalling (Leung *et al*, 2002; Lustig *et al*, 2002; Yu *et al*, 2005). In the head of a pancreas injured by PDL, where there is still healthy tissue, the reporter was inactive (Figure 1A), in agreement with the previous observations made with the TOPGAL Wnt reporter mice (DasGupta and Fuchs, 1999; Pasca di Magliano *et al*, 2007). However, after controlled injury by PDL (Watanabe *et al*, 1995; Xu *et al*, 2008), the *Axin2*$^{LacZ}$ reporter was highly activated along the ductal tree of the ligated part of the pancreas (Figure 1B). *Axin2* activation in the pancreas was already detectable at day 3 post injury, as assessed by qPCR (Figure 1C). Co-labelling with duct (pancytokeratin, CK) and endocrine (insulin, INS) markers revealed that the *Axin2* upregulation was restricted to the duct compartment (Figure 1D). Thus, pancreas injury by PDL led to activation of Wnt target genes in the proliferative duct cell compartment (Scoggins *et al*, 2000) during the regenerative response.

We have recently described that the Wnt target *Lgr5* not only marks stem cells during physiological self-renewal (e.g., in the gut), but also marks a population of liver stem cells that is activated after liver damage (Huch *et al*, 2013). We utilized the *Lgr5-LacZ* knock-in allele (Barker, *et al*, 2007) to determine the expression of the Wnt target *Lgr5* in the pancreas. *Lgr5* is essentially undetectable in the head of a pancreas injured by PDL (non-ligated pancreas), in agreement with the absence of Wnt signalling in the tissue under homeostatic conditions (Figure 1E). However, in the tail of the pancreas upon PDL, we observed a significant *Lgr5*$^{LacZ}$ reporter activity in the duct cells of the ligated pancreas, starting at day 3 and peaking at day 7 after PDL (Figure 1C and F). No background staining was detected in wild-type mice following pancreas injury (Supplementary Figure S1). The appearance of *de novo* expression of *Lgr5* following pancreas regeneration by PDL suggested that pancreatic *Lgr5* expression may herald *de novo* activation of regenerative stem/progenitor cells by Wnt upon injury.

### Pancreatic ducts self-renew in vitro

Given the induction of Wnt and *Lgr5* after injury, and the existence of pancreas progenitors in the ductal tree (Criscimanna *et al*, 2011; Furuyama *et al*, 2011), we reasoned that adult pancreas progenitors could be expanded from the duct cell compartment under our previously defined gut and stomach organoid culture conditions (Sato *et al*, 2009; Barker *et al*, 2010). Cultures of heterogeneous populations of pancreas cells have been previously established and typically include factors such as EGF, HGF and Nicotinamide (Bonner-Weir *et al*, 2000; Ramiya *et al*, 2000; Deutsch *et al*, 2001; Seaberg *et al*, 2004; Rovira *et al*, 2010; Cardinale *et al*, 2011; Smukler *et al*, 2011). Most of these approaches yield cell populations that undergo senescence over time unless the cells are transformed.

To establish pancreas cultures, isolated pancreatic duct fragments from adult healthy mice (Figure 2A) were embedded in Matrigel containing the 'generic' organoid culture factors EGF, RSPO1 and Noggin (Sato *et al*, 2009) to which FGF10 (Bhushan *et al*, 2001) and Nicotinamide were added. Under these conditions, small duct fragments formed closed structures within 24–48 h that expanded into budding cyst-like organoids (Figure 2B). The efficiency of cyst formation from isolated ducts and subsequent organoid formation was nearly 100%. Without EGF, RSPO1 or FGF10, the cultures

deteriorated after 2–5 weeks (Supplementary Figure S2A). Noggin and Nicotinamide proved to be essential to maintain the cultures >2 months (~passage 8) (Supplementary Figure S2A). The cultures maintained exponential growth with cell doubling times essentially unchanged during the culturing period (Figure 2C). Using these culture conditions, we have been able to expand the cultures by passaging at a 1:4–1:5 ratio weekly for over 10 months (Figure 2B). These culture conditions allowed the recovery of the cells after freezing and thawing. Of note, when transplanted into immunocompromised mice, the cultures did give rise only to ductal structures, and no tumour formation was detected in any of the mice analysed (*n* = 5), confirming the non-transformed origin of the cultured cells (Supplementary Figure S2C and D). Also, the karyotype analysis revealed that chromosome numbers were essentially normal, even after >5 months in culture (Supplementary Figure S2B).

Organoids generated from *Axin2*$^{LacZ}$ and *Lgr5*$^{LacZ}$ knock-in mice allowed localization of the *Axin2*- and *Lgr5*-positive cells. We observed XGAL staining in *Axin2*$^{LacZ}$ pancreas organoids throughout the cysts, whereas XGAL staining in the *Lgr5*$^{LacZ}$-derived pancreas organoids was mainly restricted to small budding structures (Figure 2D). These results resembled the *in vivo* situation after pancreas injury by PDL, where only the ductal buds were *Lgr5*$^+$, whereas the *Axin2* reporter showed a broader expression pattern (compare Figure 1B versus Figure 1F).

### Prospectively isolated single pancreatic duct cells but not endocrine or acinar cells self-renew long term in vitro

We then prospectively isolated the different pancreatic epithelial cells (duct, acinar and endocrine lineages) and cultured the different populations in our defined 3D culture system. A prospective isolation procedure that allows isolation of single cells of the different pancreatic epithelial cell types and maintenance of their viability in culture has not been established yet. The epithelial cell-surface marker EpCAM and the high concentration of $Zn^{2+}$ in secretory granules of endocrine cells, that allows binding of the fluorescent chelator TSQ (6-methoxy-8-*p*-toluenesulfonamido-quilone), were used as a basis for cell isolation. Pancreas tissue from both WT or transgenic mice that constitutively and ubiquitously express eGFP (Okabe *et al*, 1997) was dissociated into single cells. After depletion of non-epithelial (EpCAM$^-$) and haematopoietic cells (CD45$^+$, CD31$^+$), the cell suspension was FACS sorted in order to separate the granulated endocrine fraction (EpCAM$^+$TSQ$^+$) from the non-endocrine component (EpCAM$^+$TSQ$^-$) with high purity (>99.6%) (Figure 3A–D; Supplementary Figure S3A and B). To rule out the possibility that endocrine cells might de-granulate during the isolation procedure and thus contaminate the non-endocrine fraction, we repeated the protocol on pancreas cells obtained from mouse insulin promoter (Mip)-RFP mice and found no RFP$^+$ cells in the non-endocrine fraction (Supplementary Figure S4A). The separated fractions were then tested for their ability to survive, proliferate and give rise to organoids under the above-defined conditions. Only the EpCAM$^+$TSQ$^-$ exocrine cells were able to generate duct-like structures that gave rise to larger organoids (1–1.5% organoid formation efficiency) and had to be split once a week (Figure 3E). As expected, the

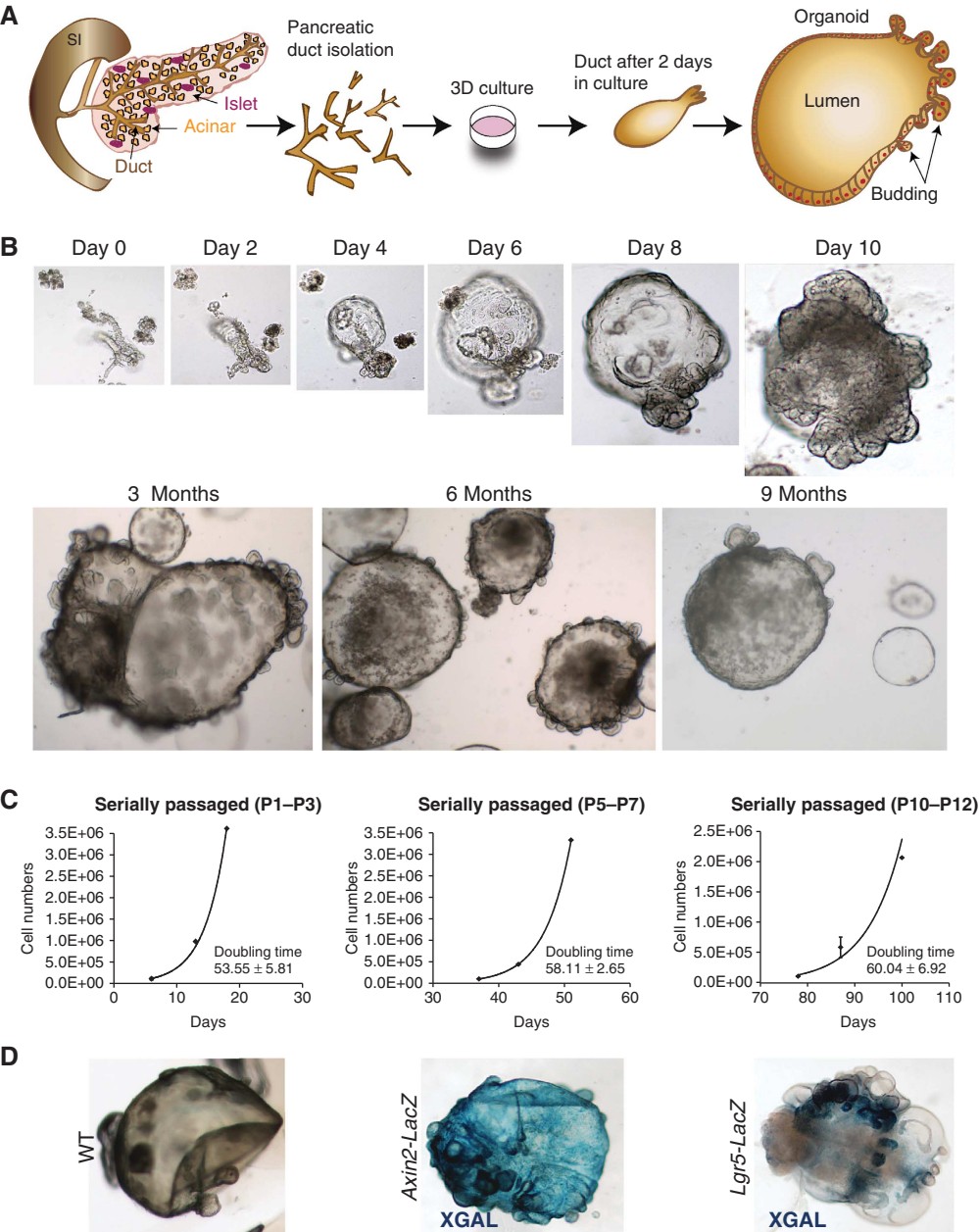

**Figure 2** Establishment of the pancreas organoids from adult pancreatic ducts. (**A**) Scheme representing the isolation method of the pancreatic ducts and the establishment of the pancreatic organoid culture. The pancreatic ducts were isolated from adult mouse pancreas after digestion, handpicked manually and embedded in matrigel. Twenty-four hours after, the pancreatic ducts closed and generated cystic structures. After several days in culture, the cystic structures started folding and budding. (**B**) Representative serial DIC images of a pancreatic organoid culture growing at the indicated time points. Magnifications: × 10 (days 0, 2, 4, 6, and 8) and × 4 (day 10 onwards). (**C**) Growth curves of pancreas cultures originated from isolated pancreatic ducts cultured as described in Materials and methods. Note that the cultures followed an exponential growth curve within each time window analysed. Graphs illustrate the number of cells counted per well at each passage from passages P1–P3 (left), P5–P7 (middle) and P10–P12 (right). The doubling time (hours) is indicated in each graph. Data represent mean ± s.e.m., $n = 2$. (**D**) Representative DIC images of XGAL staining in WT (left), *Axin2-LacZ* (middle) and *Lgr5-LacZ* (right) derived pancreas organoids.

growth pattern of the single sorted cells followed an exponential curve (Supplementary Figure S3D). The duct-derived cell cultures were maintained for > 5 months (Figure 3E). The EpCAM$^+$TSQ$^+$ endocrine cells did not proliferate, but survived for at least 30 days in culture (Figure 3F).

Acino-ductal metaplasia can happen under conditions of stress or following injury (Means *et al*, 2005; Blaine *et al*, 2010). To confirm that duct rather than acinar cells are the long-term expanded cells isolated from the EpCAM$^+$TSQ$^-$ fraction, we traced the progeny of isolated duct (*Sox9$^+$*) or

acinar (*Ptf1a$^+$*) cells *in vitro*. Transgenic mice with a *Ptf1a$^{CreER}$* allele, that is exclusively expressed in the acinar compartment (Kopp *et al*, 2012; Pan *et al*, 2013), or mice carrying the *Sox9$^{CreER}$* allele, that is expressed predominantly (but is not absolutely restricted to) the duct cell compartment (Furuyama, *et al*, 2011; Kopp *et al*, 2012) were crossed with *Rosa26R$^{YFP}$* mice and subcutaneously injected with tamoxifen as described in Supplementary Figure S5A. After the washout period, the pancreas was dissociated and single *Sox9$^{YFP+}$* or *Ptf1a$^{YFP+}$* cells were FACS sorted and cultured in our defined pancreas

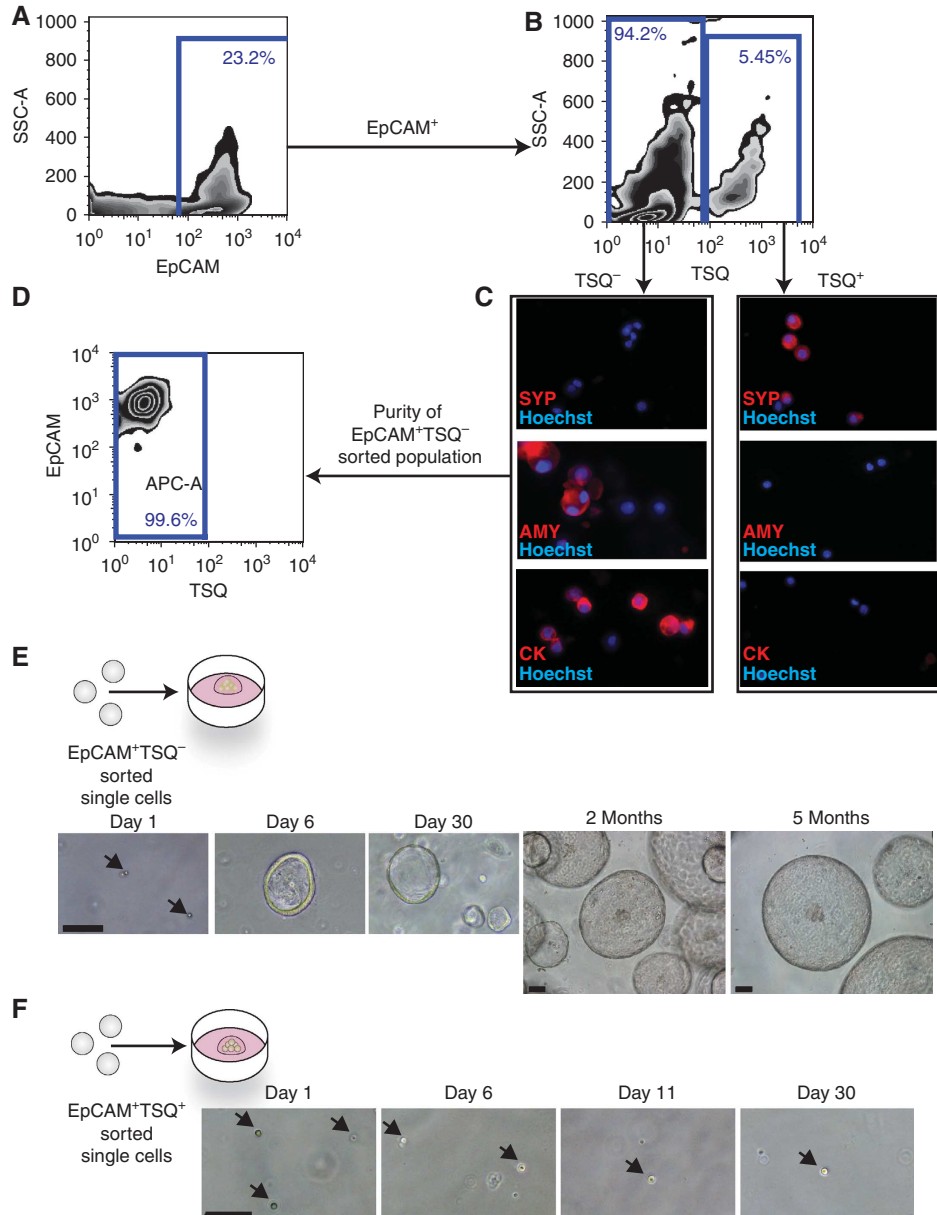

**Figure 3** Isolation and *in vitro* expansion of single, endocrine-depleted pancreatic epithelial cells. (**A**) Representative fluorescence-activated cell sorting (FACS) plot illustrating the distribution of EpCAM$^+$ and EpCAM$^-$ cells from dissociated adult mouse pancreas, following epithelial cell enrichment by magnetic beads as described in Materials and methods. (**B**) Representative FACS plot showing the distribution of EpCAM$^+$ non-granulated TSQ$^-$ epithelial cells and EpCAM$^+$ granulated TSQ$^+$ endocrine cells. (**C**) EpCAM$^+$TSQ$^+$ and EpCAM$^+$TSQ$^-$ sorted fractions were cytocentrifuged and immunostained (red) for Synaptophysin (SYP), Amylase (AMY) and pancytokeratin (CK); nuclei were counterstained with Hoechst 33342 (blue). Magnification: $\times$ 40. (**D**) Representative FACS analysis purity of sorted EpCAM$^+$TSQ$^-$ cells indicating that this population is isolated with high purity ($>$99.6%). (**E**) EpCAM$^+$TSQ$^-$-sorted single cells were assessed for their growth potential in 3D expansion culture conditions: this population gave rise to organoids that could be expanded for many passages ($>$5 months). (**F**) EpCAM$^+$TSQ$^+$-sorted single cells were assessed for their growth potential under the same conditions: endocrine TSQ$^+$ cells survived in culture but did not proliferate. Scale bars: 30 μm.

culture medium (Supplementary Figure S5B–D). Only *Sox9*$^{YFP+}$ cells grew into budding organoids that expanded long term in culture, even when starting from a single cell (Supplementary Figure S5D, top panel). By contrast, the cultures derived from *Ptf1a*$^{YFP+}$ cells gave rise to smaller duct-like structures that were able to proliferate only for 3–4 passages, after which they arrested proliferation (Supplementary Figure S5D, bottom panel). In conclusion, these data indicated that the long-term expanding pancreas organoid cultures derive from duct cells.

### Lgr5 cells sustain the growth of pancreas organoids that have a duct cell phenotype

To test whether the *Lgr5*-expressing cells maintained the growth potential of the pancreas organoids, we sorted single *Lgr5*$^{LacZ+}$ cells from *in vitro* expanded organoids derived from *Lgr5-LacZ* knock-in mice (Barker *et al*, 2007). Indeed, the isolated *Lgr5*$^+$ cells grew and formed organoids (Figure 4A–E) that were subsequently expanded for $>$4 months in culture by splitting the cultures weekly at a 1:6–1:8 ratio. The colony formation efficiency was $\sim$16%, similar

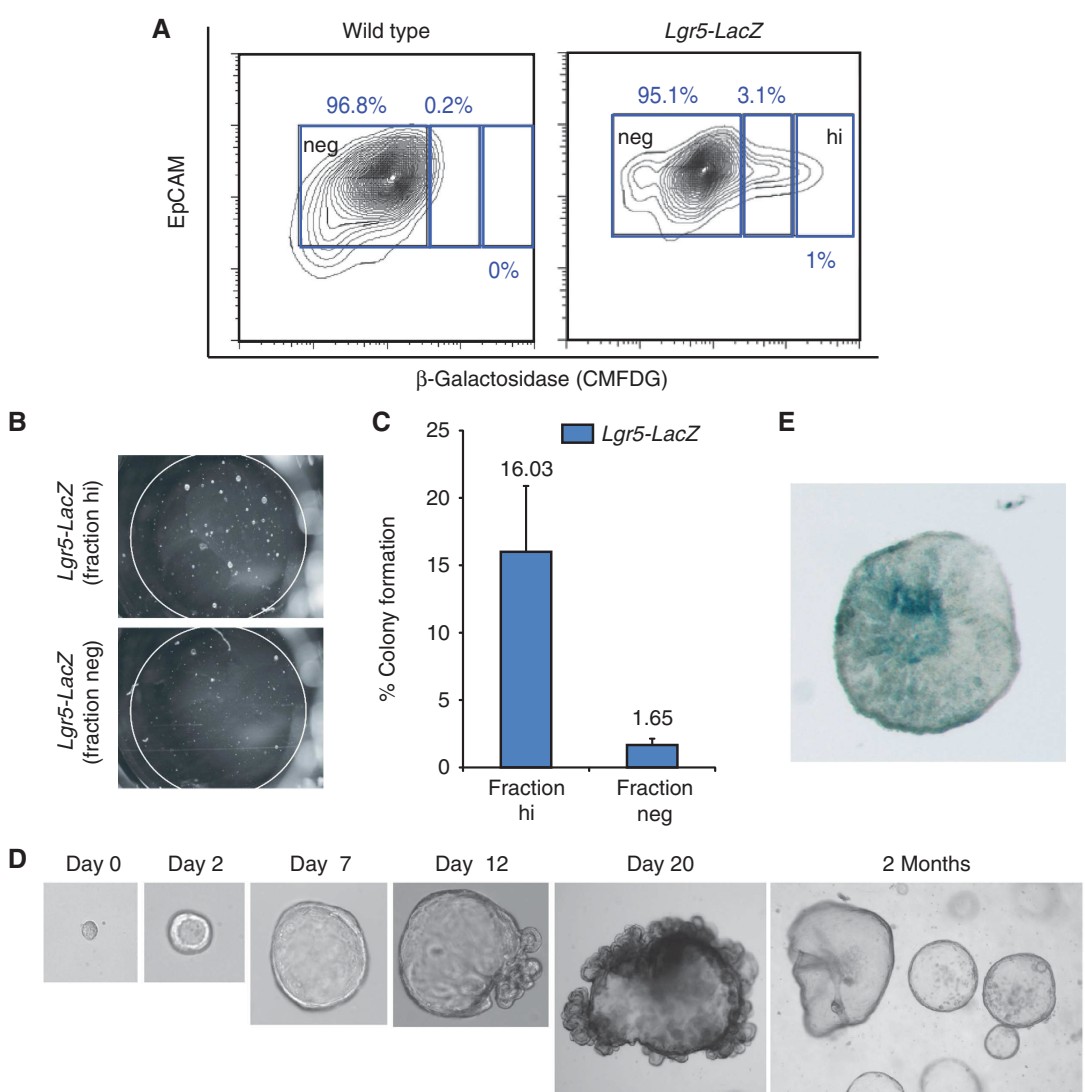

**Figure 4** Clonal expansion of single *Lgr5* cells derived from *Lgr5-LacZ* pancreatic organoids. (**A–E**) Lgr5[+] cells were sorted from *Lgr5[LacZ]*-derived pancreas organoids cultured for 20–25 days in our defined medium. Organoid formation efficiency was analysed 12 days after seeding. (**A**) Representative image of an FACS plot of wild-type (left) and *Lgr5[LacZ]* (right) pancreas organoids stained with Detectagene Green CMFDG (for detecting beta-galactosidase expression) and EpCAM (for selecting epithelial cells). (**B**) Representative image of cultures derived from 500 sorted cells from the high (*Lgr5[hi]*) or 500 sorted cells derived from the negative fraction (*Lgr5[neg]*) 12 days after seeding. (**C**) Graph showing the % of colony formation efficiency of the *Lgr5[hi]* and *Lgr5[neg]* fractions. (**D**) Representative serial DIC images showing the outgrowth of pancreatic organoids originated from a single *Lgr5[LacZ+]* cell. Magnifications: ×40 (days 1–2), ×20 (day 7), ×10 (days 12–20), ×4 (2 months). (**E**) Representative DIC image of XGAL staining in a 12-day-old clonal culture derived from *Lgr5[hi]* fraction. Magnification: ×10.

to the colony formation of Lgr5 cells of small intestine and stomach (Barker *et al*, 2010; Sato *et al*, 2011) (Figure 4C). Of note, 1.6% of the *Lgr5[neg]*-sorted population also grew into organoids (Figure 4C; Supplementary Figure S6A–D). These *Lgr5[neg]*-derived clones rapidly re-expressed *Lgr5* (Supplementary Figure S6B and C) and expanded at a similar ratio as their *Lgr5[+]* counterparts (Supplementary Figure S6D). This result mirrors the efficiency of colony formation of the FACS-sorted EpCAM[+]TSQ[−] exocrine cells from healthy tissue (1.65%, *Lgr5[neg]* versus 1–1.5%, exocrine cells). Overall, these results demonstrated that pancreas-derived *Lgr5[+]* cells are capable of self-renewal and expansion *in vitro,* indicating that stem/progenitor cells can be activated both in organ-like structures and in secondary, single cell-derived organoids.

Organoids derived from single, FACS-sorted, *Sox9[+]* duct cells or from single isolated *Lgr5[+]* cells (FACS sorted from *Lgr5[LacZ]* cultures) allowed us to assess their lineage potential *in vitro*. Histologically, pancreas organoids displayed a duct-like phenotype characterized by a single-layered epithelium of cytokeratin-positive (CK) and MIC1-1C3-positive (Dorrell *et al*, 2008) cells (Figure 5A). *Lgr5[+]* cells were readily detected in all organoids analysed (Figure 4E), similarly to what we had observed in the cultures derived from (non-clonal) duct fragments (Figure 2D). Ki67 and Edu staining demonstrated that only a subset of cells within the organoids proliferate (Figure 5A).

Then, we performed comparative gene expression profiling of 1- to 2-month-old cultures and compared it with the gene expression profile of adult duct, acinar and islet cells. The

overall gene expression profile of the organoid cultures clustered with the duct cell arrays, whereas it did not cluster with the gene profiles of acinar or endocrine cells (Figure 5B). Of note, among the genes whose expression pattern did cluster between the duct pancreatic cells and the organoids we found *Sox9, Krt7, Krt19* and *Spp1* (full list is provided in Supplementary Dataset 1). Comparison of the gene expression profile of the pancreas organoids and the pancreatic tissue (by *in silico* subtraction) confirmed the segregation of the non-ductal pancreatic markers (like *Sst, Ins2, Gcg* and *Amy*) and

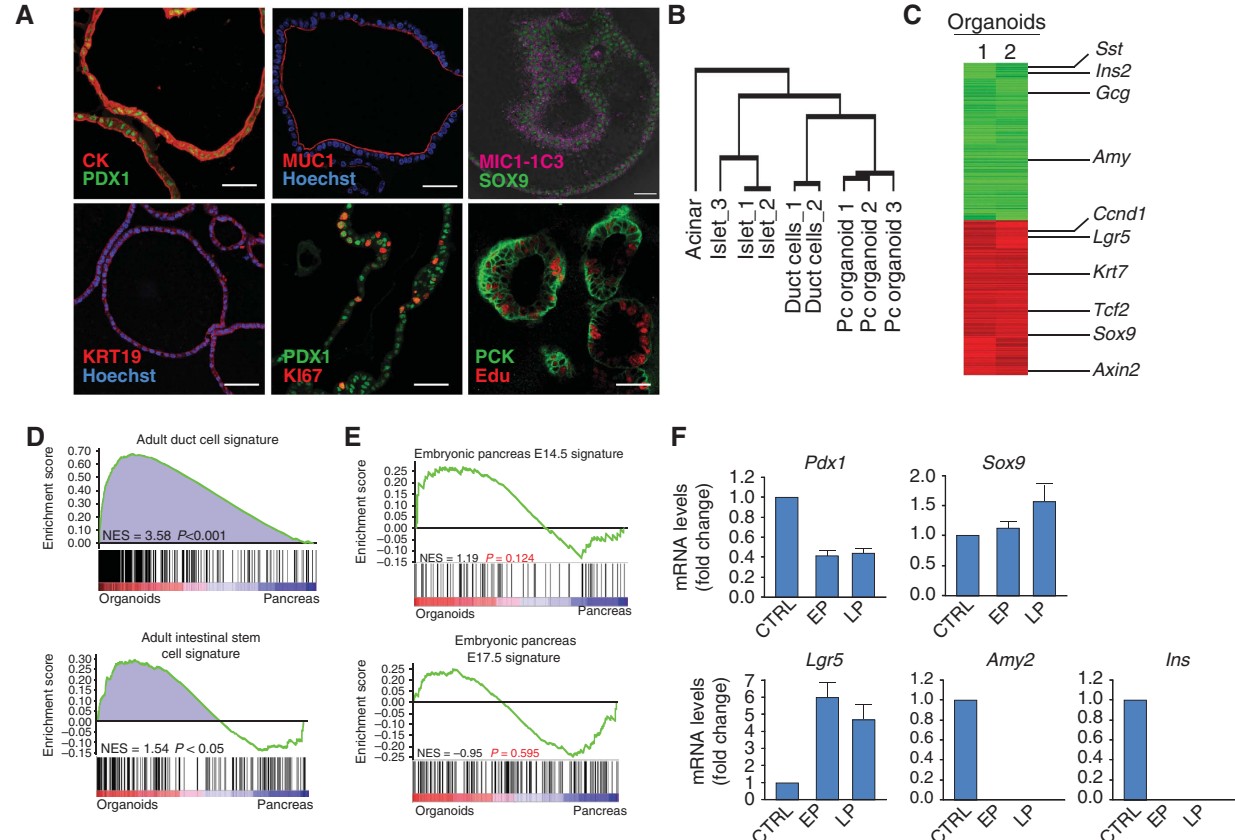

**Figure 5** Characterization of *in vitro* expanded organoids from single cells. Pancreatic organoids cultured in our defined medium show a ductal-like phenotype. FACS-sorted epithelial cells were isolated as shown in Figures 3 and 4, plated in matrigel and cultured for 7–9 passages before being processed for RNA and immunohistochemistry analysis. (**A**) Organoids were stained with anti-pancytokeratin (CK), anti-PDX1, anti-mucin 1 (MUC1), anti-SOX9, anti-cytokeratin19 (CK19) and anti-MIC1-1C3 that revealed their duct-like phenotype. Cells were proliferating, as illustrated by anti-Ki67 and Edu immunostaining. Scale bars: 35 μm. (**B**) Hierarchical clustering analysis of genes differentially expressed among pancreatic organoid cultures, duct, acinar and islet pancreatic lineages. Unsupervised hierarchical clustering analysis shows that the pancreas organoid arrays cluster with the ductal array while the acinar and islet profiles clustered in a separate tree. (**C**) Heat map of two independent clonal pancreas organoid cultures performed after subtracting pancreas tissue expression levels. Representative organoid-, duct-, endocrine- and exocrine-specific genes are listed on the right. Red, upregulated; green, downregulated. (**D, E**) Gene set enrichment analysis (GSEA) revealed that pancreas organoid cultures express a significant amount of genes contained in signatures of adult duct cells and intestinal stem cells (Muñoz *et al*, 2012) (**D**), while they were not enriched in gene sets representing embryonic pancreas at E14.5 or E17.5 (Juhl *et al*, 2008) (**E**). (**F**) Quantitative PCR showing relative fold changes of *Pdx1, Sox9, Lgr5, Amy2* and *Ins* mRNA levels in partial duct ligated pancreas (CTRL) and cultured organoids at early (EP, passages 2–3) and late passages (LP, passages 7–9). *Cyclophillin A* was used to normalize for RNA input. Data represent mean ± s.e.m. ($n = 2$, independent cultures).

**Figure 6** *In vitro* expanded organoids from single epithelial cells give rise to endocrine and duct cells when grafted *in vivo* in a developing pancreas. (**A–F**) Pancreas organoid cultures were derived from CAG$^{eGFP+}$ mice or ECad$^{CFP+}$ mice as described in Figure 3 and Supplementary Figure S3. The cultures were clonally expanded *in vitro* for 4–6 passages before dissociation into single cells. Dissociated eGFP$^+$ or ECad$^{CFP+}$ cells were re-aggregated with WT embryonic E13 mouse (**B, C**) or E14 rat (**D, E**) pancreas. The re-aggregates were kept on a filter membrane O/N and then grafted under the kidney capsule of nude mice. The grafts were harvested and analysed 2–3 weeks after. The re-aggregates consistently grew and gave rise to pancreatic tissue, as illustrated in Supplementary Figure S7A. (**A**) Schematic representation of the pancreatic morphogenetic assay. (**B**) Representative confocal microscopy image showing incorporation of eGFP$^+$ cells (green) into pancytokeratin$^+$ (CK, red) pancreatic duct structures; these eGFP$^+$ cells also express low levels of PDX1 (blue). Other eGFP$^+$ cells (white arrow) aggregated in islet-like structures near the ducts, downregulated CK and expressed high levels of PDX1 (blue). Scale bar: 35 μm. (**C**) Confocal microscopy demonstrates that cultured eGFP$^+$ (green) cells differentiate into beta cells and express both synaptophysin (SYP, red) and insulin (INS, blue). Scale bar: 20 μm. (**D**) Confocal microscopy image illustrating mouse Insulin$^+$ Cpeptide$^+$ (INS$^+$Cppt$^+$) cells derived from ECad$^{CFP+}$-grafted cells. Note that the INS$^+$Cppt$^+$ cells are incorporated into an embryonic rat pancreas, where rat INS$^+$ cells are negative for mouse-specific Cppt staining (dotted line). Scale bar: 30 μm. (**E**) High magnification of an ECad$^+$INS$^+$Cppt$^+$-grafted cell (**D**) that displays CFP membrane localization and cytoplasmic staining for INS and mouse Cppt. Scale bar: 20 μm. (**F**) Histogram showing the average quantification of differentiation of eGFP$^+$ cells engrafted into each re-aggregate under the kidney capsule. Endocrine (SYP$^+$), Insulin (INS$^+$), duct cells (CK$^+$); others: cells expressing neither duct nor endocrine markers. Average number/graft ± s.e.m. $n = 11$ grafts from six independent cultures.

the ductal markers (like *Krt7, Tcf2* and *Sox9*) (Figure 5C; Supplementary Dataset 2). Of note, the Wnt target genes *Lgr5, Ccnd1* and *Axin2* were also specifically highly expressed in the organoids (Figure 5C; Supplementary Dataset 2). As expected, gene set enrichment analysis (GSEA) confirmed that the organoid cultures are enriched in genes specifically expressed in adult Sox9$^+$ pancreatic duct cells (Figure 5D

and Supplementary Dataset 3). Interestingly, we also observed enrichment in genes previously reported in small intestinal and pancreas stem cells, that is, *Lgr5, Prom1, Sox9* and *Lrig1* (Figure 5D and Supplementary Dataset 4) (Barker *et al*, 2007; Snippert *et al*, 2009, Furuyama *et al*, 2011; Wong *et al*, 2012), while we found no significant enrichment in genes expressed in the developing pancreas at E14.5 or E17.5

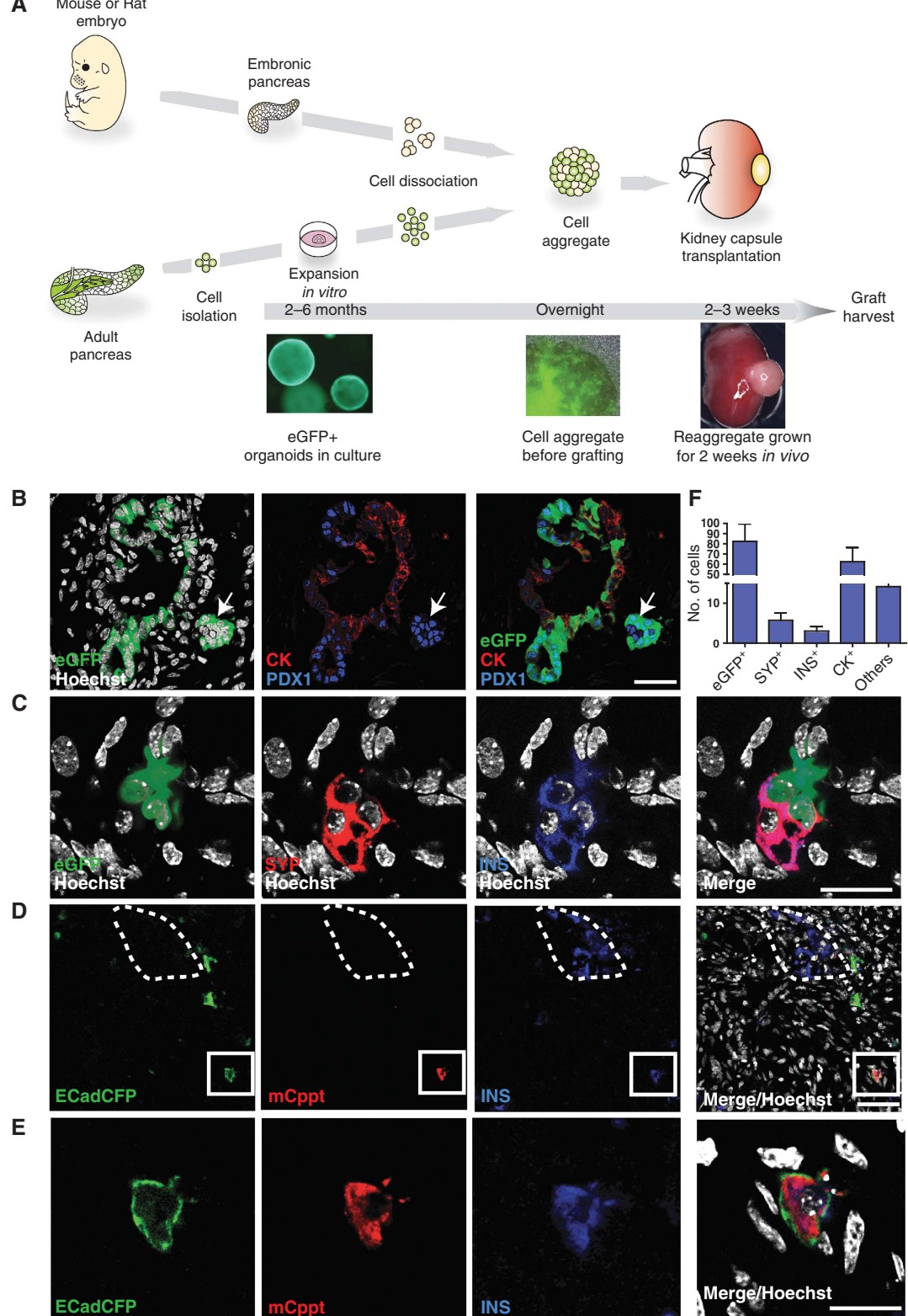

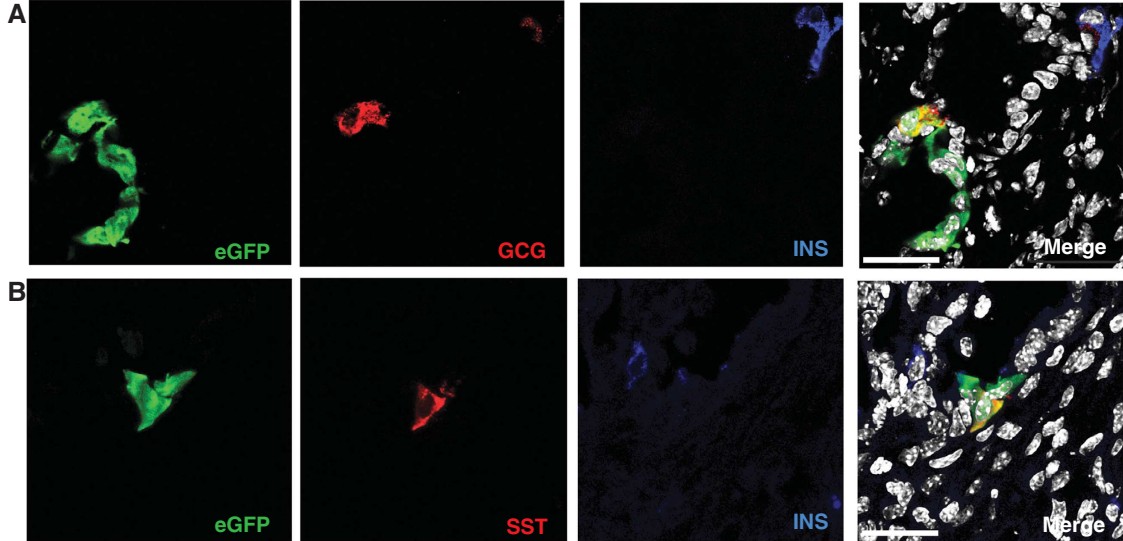

**Figure 7** *In vitro* expanded organoids give rise to Glucagon (GCG) and Somatostatin (SST) mono-hormonal cells *in vivo*. Pancreas organoid cultures derived from sorted eGFP$^+$ cells and expanded for at least 2 months in culture were grafted as described in Figure 6. Two weeks after transplantation, the kidney grafts were harvested and the grafted eGFP$^+$ cells were evaluated for the expression of glucagon (GCG) and Somatostatin (SST). (**A**, **B**) Representative images illustrating that eGFP$^+$ cells (green) differentiate towards GCG$^+$ (**A**) or SST$^+$ (**B**) cells *in vivo*. Note that both Gcg$^+$ and SST$^+$ cells do not express INS (insulin, blue), indicating that the organoid-derived eGFP$^+$ cells have fully differentiated into mono-hormonal cells *in vivo*. Scale bars = 20 μm.

(Figure 5E and Supplementary Datasets 5 and 6), confirming the adult nature of our pancreas progenitor cultures. To confirm this expression pattern, we performed qPCR analysis in cultures at early and late passages (Figure 5F). While some genes could be detected in pancreas organoids over time (*Pdx1*, *Sox9* and *Lgr5*), no acinar (*Amy2*) or endocrine (*Ins*) markers were observed over several passages (Figure 5F). Immunofluorescent staining confirmed that the organoids were mainly formed by cells expressing Keratin19 (KRT19), SOX9, MUCIN-1 and PDX1 (Figure 5A) while negative for the endocrine marker Synaptophysin (SYP) (Supplementary Figure S4B). Overall, these results confirmed the pancreas progenitor and duct-like nature of the pancreas organoid cultures.

### Expanded organoids give rise to both pancreatic endocrine and duct cells in vivo

The embryonic pancreas harbours all necessary factors and appropriate environmental cues to support the differentiation of *bona fide* pancreas progenitors to mature exocrine and endocrine cells *in situ* or when the embryonic pancreas is transplanted under the kidney capsule of an immunodeficient mouse (Zaret and Grompe, 2008). Therefore, to assess whether the organoid cells are capable of differentiating towards fully mature endocrine lineages (e.g., insulin pro-ducing cells) we developed a whole-organ morphogenetic assay based on the re-aggregation of dissociated cells from embryonic pancreas on one hand and organoids generated from adult pancreas on the other hand (Figure 6A; Supplementary Figure S7B). This type of morphogenetic assay has successfully been used to demonstrate fate potency of both skin and thymic epithelial stem cells after expansion *in vitro* (Bonfanti *et al*, 2010). When embryonic pancreas derived from either mouse (E13.5) or rat (E14) was isolated, dissociated, re-aggregated and then transplanted under the kidney capsule of an immune-deficient mouse, the embryonic

tissue fully developed into the three mature pancreas lineages: duct, acinar and endocrine cells (Supplementary Figure S7A).

Therefore, we isolated EpCAM$^+$ TSQ$^-$ GFP$^+$ epithelial cells from the pancreas of CAG$^{eGFP}$ adult mice (Okabe *et al*, 1997) as described above and expanded for at least 6 weeks (Supplementary Figure S3A–C), dissociated them into a single cell suspension and re-aggregated with embryonic E13 or E14 WT mouse or rat pancreas, respectively. The re-aggregates were kept overnight on a membrane and, the day after, were grafted under the kidney capsule of nude mice. After 2 or 3 weeks, mice were sacrificed and grafts harvested (Figure 6A; Supplementary Figure S7B). The transplanted re-aggregates did consistently grow pancreatic structures organized in both exocrine and endocrine areas, several of which contained eGFP$^+$ integrated cells (Figure 6B and C; Supplementary Figure S7C and D). Immunohistochemical analysis of the re-aggregates revealed that eGFP$^+$ cells mainly contributed to duct cells (Figure 6B and F). Of note, some eGFP$^+$ cells, that located outside of the ducts, downregulated cytokeratin expression and contained high level of PDX1 protein, a feature of beta cells (Figure 6B). On the basis of their expression of synaptophysin, ~5% of integrated eGFP$^+$ cells were of endocrine nature, 50% of which were also insulin$^+$ (Figure 6C and F). Quantification revealed that eGFP$^+$ cells differentiated into duct cells at a frequency of 70% (Figure 6F; Supplementary Figure S7F). It is important to remark that these percentages roughly correspond to those found in the differentiating embryonic pancreas *in vivo*. More importantly, we obtained the same results using a different reporter mouse that expresses CFP under the control of E-Cadherin promoter (ECad$^{CFP}$) (Figure 6D and E; Supplementary Figures S7E and S8). We found that INS$^+$ cells derived from cultivated organoids either from CAG$^{eGFP}$ or from ECad$^{CFP}$ reporter mice were functional and expressed C-peptide (Cppt) protein

(Figure 6D and E; Supplementary Figure S7E). Cells with CFP membrane localization and cytoplasmic expression of both INS and mouse-specific C-peptide were readily detected throughout the grafted area, even when the organoid cells were engrafted in a rat pancreas microenvironment, where endogeneous INS$^+$ cells were negative for the mouse/specific anti-Cppt antibody (Figure 6D and E; Supplementary Figure S8C). This last result excluded the possibility of fusion between mouse-cultivated eGFP$^+$ or CFP$^+$ cells and WT rat endocrine cells. The specificity of this antibody both in the ectopic rat pancreas and in the adult rat pancreas is shown in Figure 6D and Supplementary Figure S8B and C. Furthermore, the cultivated eGFP$^+$ cells also gave rise to other endocrine lineages, such as Glucagon$^+$ (GCG$^+$) and Somatostatin$^+$ (SST$^+$) cells (Figure 7A and B). These cells were negative for INS, demonstrating that they had fully differentiated into mono-hormonal endocrine cells (Figure 7A and B).

Then, we assessed whether a less permissive environment would also allow the adult expanded progenitor duct cells to achieve an endocrine cell fate. We directly transplanted ∼2-month-old adult duct pancreas cultures derived from both Bl6 WT mice and *Ecad*$^{CFP}$ mice into the kidney capsule of immunodeficient mice. We previously primed the 2-month-old cultures to express early endocrine markers by culturing them, 15 days prior to the transplantation, in a medium previously reported to allow ESC to acquire an endocrine fate (D'Amour *et al*, 2006; Kroon *et al*, 2008), with some modifications. We included the small molecule inhibitor ILV combined with FGF10, to induce *Pdx1* expression (Bhushan *et al*, 2001; Chen *et al*, 2009), followed by DBZ treatment, to inhibit notch signalling (Milano *et al*, 2004) (Supplementary Figure S9A). This medium facilitated the expression of early endocrine progenitor markers (*Neurogn3* and *Chga*) while retained the expression of the ductal marker *Sox9*, and suppressed *Lgr5* (Supplementary Figure S9B). One month after transplantation, duct-like structures formed by Krt19$^+$ cells were readily detectable throughout the graft (Supplementary Figure S9C). Also, albeit at much lower efficiency, Insulin$^+$ and Cpeptide$^+$ cells (Supplementary Figure S9C, D and E), as well as ChgA$^+$ cells were detected (Supplementary Figure S9F).

Overall, these results conclusively demonstrate that cultured organoids derived from either sorted adult duct cells (*CAG*$^{eGFP}$ or *ECad*$^{CFP}$ mice) or from freshly isolated ducts (Bl6 WT mice) are able to acquire both duct and endocrine fates, thus demonstrating their progenitor nature and bi-potency.

## Discussion

The pancreas is a glandular organ that serves two important functions: the production of the digestive enzymes and the production of the hormones responsible of glucose homeostasis. This is mirrored in the wide range of pancreas diseases that vary from pancreatic cancer to disorders related to the glucose homeostasis, such as diabetes. While pancreas cancer is the result of the accumulation of oncogenic mutations in different epithelial cell types of the pancreas, diabetes is the result of severe reduction in functional beta-cell mass. The lack of primary culture systems capable of long-term expansion of primary tissue *in vitro* hampers the development of therapeutic strategies for pancreas diseases. The replacement of functional pancreatic beta cells may be envisioned as a potential definitive cure for diabetes. Unfortunately, human islet transplantation is hampered by the scarcity of donors and the need for immune suppression and also by graft failure (Lysy *et al*, 2012). Therefore, alternative sources for cell therapy replacement hold promise as a potential treatment for diabetes.

ESCs and iPSCs can be differentiated towards beta cells *in vitro* (D'Amour *et al*, 2006; Zhang *et al*, 2009; Nostro *et al*, 2011; Cheng *et al*, 2012) and *in vivo* (Soria *et al*, 2000; Kroon *et al*, 2008; Sneddon *et al*, 2012), but the reproducibility of such procedures has been limited (Lysy *et al*, 2012). In addition, undifferentiated ESCs and iPSCs are prone to form teratomas upon transplantation *in vivo*, therefore any remaining undifferentiated cell must be completely removed prior to be used for transplantation. Adult pancreas progenitors able to expand long term *in vitro* while maintaining the potency to differentiate towards a duct or endocrine fate would potentially not encounter these limitations.

We report here that damage of adult pancreas results in the upregulation of Wnt signalling and expression of the stem-cell marker *Lgr5* in the neo-formed ducts. We exploit this Wnt-driven regenerative response to define a culture medium based on the Wnt activation (RSPO1) that allows the unlimited expansion of duct fragments or even single isolated cells in a defined medium without serum. Under these conditions, pancreatic duct cells upregulate the stem-cell marker *Lgr5* (receptor for RSPO1), and self-renew while maintaining their genetic stability. Importantly, when the expanded adult progenitor cells receive the appropriate differentiation signals, as for instance the signals present in a developing embryonic pancreas, they are able to integrate into both exocrine and endocrine structures that express functional markers, demonstrating that they carry the hallmarks of bi-potent progenitors.

Confirming the importance of the Wnt/Rspo signalling to facilitate the proliferation of pancreatic adult cells, Jin *et al* reported (while this study was under revision) that Rspo supplementation to a 3-week pancreatic culture facilitates the expansion of pancreas cells into heterogeneous cultures. The otherwise non-defined medium contains fetal bovine serum and ESC-derived conditioned medium (Jin *et al*, 2013).

Thus, the conditions here described, based on the induction of the Wnt-Lgr5-Rspo axis, allow the long-term *in vitro* expansion of pancreas progenitors. The unlimited expansion potential of the adult progenitor cells may open avenues for building patient-derived disease models, as well as the development of regenerative strategies based on the expansion of adult, genetically non-modified, pancreas cells. Future optimization of the differentiation conditions may allow the generation of high numbers of specialized and functional pancreatic cells to be used for the treatment of pancreas diseases such as diabetes.

## Materials and methods

### Mice lines and injury models
Generation and genotyping of the *Lgr5*$^{LacZ}$ and *ECad*$^{CFP}$ mice is already described in Barker *et al* (2007) and Snippert *et al* (2010), respectively. *Axin2*$^{LacZ}$ mice were obtained from EMMA (European Mouse Mutant Archive, Germany). C57BL/6-Tg(ACTB-EGFP)1Osb/J, *Sox9*$^{CreER}$ and *Ptf1a*$^{CreER}$ mice were previously described (Okabe *et al*,

1997; Furuyama *et al*, 2011; Pan *et al*, 2013) and MipRFP mice were provided by Gérard Gradwohl (IGBMC, Strasbourg, France). Wild-type Sprague–Dawley (OFA) rats and OF1 mice were obtained from Janvier. Athymic (Swiss Nu$^{-/-}$) were supplied by Charles River Breeding Laboratories NSG mice (Jackson Laboratory, Bar Harbor, MA, USA). All animal experiments were performed in accordance with the institutional review committee at the Hubrecht Institute and the VUB. Animals were maintained in a 12-h light cycle providing food and water *ad libitum*.

To induce pancreas injury, 3- to 6-month-old mice were anaesthetized, with a mixture of fluanisone:fentanyl:midazolam injected intraperitoneally at a dosage of 3.3, 0.105 and 1.25 mg/kg, respectively. Following a median incision on the abdominal wall, the pancreas was exposed and, under a dissecting microscope, the pancreatic duct was ligated as described (Xu *et al*, 2008).

For Sox9 and Ptf1a lineage labelling, tamoxifen (Sigma, T5648) was prepared at the concentration of 10 mg/ml in corn oil (Sigma, C8267). A total dose of 20 mg of tamoxifen was given subcutaneously in five doses of 4 mg over a 10-day period. A washout period of 14 days preceded pancreas harvesting and dissociation into single cells.

### Pancreas organoid cell culture

Pancreatic ducts were isolated from the bulk of the pancreas of mice older than 8 weeks by collagenase dissociation (Collagenase type XI 0.012% (w/v) (Sigma), dispase 0.012% (w/v) (Gibco), FBS (Gibco) 1% in DMEM media (Gibco)) at 37°C. Isolated ducts were mixed with Matrigel (BD Bioscience) and seeded and cultured as we described previously (Sato *et al*, 2009; Barker *et al*, 2010). After Matrigel formed a gel, culture medium was added. Culture media was based on AdDMEM/F12 (Invitrogen) supplemented with B27 (Invitrogen), 1.25 mM *N*-Acetylcysteine (Sigma), 10 nM gastrin (Sigma) and the growth factors: 50 ng/ml EGF (Peprotech), 10% RSPO1-conditioned media (kindly provided by Calvin Kuo), 100 ng/ml Noggin (Peprotech) or 10% Noggin-conditioned media (in-house prepared), 100 ng/ml FGF10 (Peprotech) and 10 mM Nicotinamide (Sigma). One week after seeding, organoids were removed from the Matrigel, mechanically dissociated into small fragments, and transferred to fresh Matrigel. Passage was performed in a 1:4–1:8 split ratio once per week for at least 9 months. To prepare frozen stocks, organoid cultures were dissociated and mixed with Recovery cell culture freezing medium (Gibco) and froze following the standard procedures. When required, the cultures were thawed using standard thawing procedures, embedded in Matrigel and cultured as described above. For the first 3 days after thawing, the culture medium was supplemented with Y-27632 (10 μM, Sigma-Aldrich).

### Prospective isolation and pancreas organoid single cell (clonal) culture

For clonogenic assays, whole pancreata were harvested from adult (8–12 weeks) mice and individually digested by collagenase type XI (0.3 mg/ml, Sigma) incubation at 37°C in a shaking incubator, and then dissociated into single cells by addition of trypsin (1 mg/ml, Sigma) and DNAse (0.4 mg/ml, Roche); cell suspension was filtered through a 70-μm cell strainer. Cell pellets were incubated with anti-mouse EpCAM/APC antibody (eBiosciences) for 30′ on ice. Cells were either processed directly for FACS sorting or were enriched for epithelial cells using magnetic beads (EasySep$^{TM}$ APC Positive selection kit or Epithelial enrichment kit; STEMCELL Technologies Inc.). Cells were re-suspended in a solution containing propidium iodide (PI, 1 mg/ml, Sigma), and *N*-(6-Methoxy-8-Quinolyl)-*p*-Toluenesulfonamide (TSQ, 1 mg/ml, Molecular Probes) and sorted on an FACSAria (Becton Dickinson). Clean separation between EpCAM$^+$TSQ$^-$ and EpCAM$^+$TSQ$^+$ cell populations was confirmed by a second FACS analysis and immunocytochemistry. According to the mouse strain, an additional gate for eGFP or YFP signal was used for sorting cells. Pulse-width gating excluded cell doublets while dead cells were excluded by addition of PI and gating on the negative cells.

For secondary clonal cultures, established cultures were dissociated into single cells and stained with the DetectaGene Green CMFDG LacZ Gene Expression Kit (Molecular Probes) according to the manufacturer's instructions. PI staining was used to label dead cells and FSC: pulse-width gating to exclude cell doublets.

Sorted cells (EpCAM$^+$TSQ$^-$, EpCAM$^+$TSQ$^+$ or Lgr5$^{LacZ+}$) were embedded in Matrigel and seeded in 96-well plates at a ratio of 1 sorted cell/well. Cells were cultured in the pancreas media described above supplemented with Y-27632 (10 μM, Sigma-Aldrich) for the first 4 days. Passage was performed in split ratios of 1:4–1:5 once per week for at least 6 months.

### In vitro growth curves

Expansion ratios were calculated from both sorted cells and duct fragments as follows: pancreas organoid cultures or $20 \times 10^3$ sorted cells were grown in our defined medium for 7 days. Then, the cultures were dissociated by incubation with TrypLE Express (Gibco) until single cells. Cell numbers were counted by trypan blue exclusion at the indicated time points. From the basic formula of the exponential curve $y(t) = y_0 \times e^{(\text{growth rate} \times t)}$ ($y =$ cell numbers at final time point; $y_0 =$ cell numbers at initial time point; $t =$ time) we derived the growth rate. Then, the doubling time was calculated as doubling time $= \ln(2)/$growth rate for each time window analysed.

### Karyotyping

Organoid cultures in exponential growing phase were incubated for 1–1.5 h with 0.05 μg/ml colcemid (Gibco). Then, cultures were dissociated into single cells using TrypLE express (Gibco) and processed as described (Huch *et al*, 2013). Chromosomes from 100 metaphase-arrested cells were counted.

### Pancreatic morphogenetic assay

Pancreatic aggregates were obtained following a previously described protocol (Bonfanti *et al*, 2010), modified as follows. E13 mouse embryos (OF1) or E14 rat embryos (SD) were harvested from the uteri under sterile conditions, transferred in 100 mm Petri dishes containing HBSS supplemented with 10% FCS and stored on ice. Pancreatic tissue was removed from the embryonic abdomen and transferred into a solution containing collagenase type XI (1 mg/ml, Sigma) and DNAase (0.4 mg/ml, Roche Diagnostic) for about 5 min. A known number (from $75 \times 10^3$ to $10^5$) of GFP-labelled single cells dissociated from *in vitro* expanded adult organoids were mixed with an $\sim$10-fold excess of unlabelled embryonic pancreatic cells. Aggregates were then transferred on a 0.8-μm Isopore membrane filter (Millipore) and incubated at 37°C for 24 h in RPMI medium supplemented with 10% FCS, before being grafted under the kidney capsule of nude mice as previously described (Bonfanti *et al*, 2010). Two to four weeks later, the grafts were harvested and processed for cryosection and immunohistochemistry.

### Pancreas organoid differentiation and kidney capsule transplantation

Pancreas organoids derived from Bl6 isolated ducts or eGFP$^+$- or CFP$^+$-sorted cells were expanded *in vitro* for at least 2 months in our defined culture medium (EM) as described above. Then, the organoids were transferred into a differentiation medium (DM) to enhance their endocrine fate. To define the differentiation medium, we adapted the protocols already described by D'Amour *et al* (2006) and Chen *et al* (2009) as follows: organoids grown in Matrigel, in our defined expansion medium (EM), were removed from the Matrigel by using BD cell recovery solution (BD Biosciences), following the manufacturer's instructions, and transferred to suspension plates. The cells were maintained for 3 days in RPMI medium supplemented with 0.2% FBS and 100 ng/ml Activin A (Tocris BioScience). Then, the medium was changed to RPMI supplemented with 300 nM ILV (indolactam-V) (Tocris BioScience), 100 ng/ml FGF10 (Peprotech) and 2% FBS for 4–5 days. After, the medium was replaced by DMEM supplemented with 1% B27, Noggin (50 ng/ml), Retinoic Acid (2 μM) and KAAD-cyclopamine (0.25 μM) for the following 6 days. Finally, for the last 2–4 days prior to transplantation, the medium was changed to DMEM supplemented with 1% B27 and 10 μM DBZ (Tocris BioScience). During all the differentiation protocol, the cells were kept in suspension plates. After the last 2–4 days in DBZ supplemented medium, the organoids were collected and transplanted directly into the kidney capsule of nude mice using standard procedures. The grafts were allowed to grow for 1 month and then were harvested and processed for paraffin embedding and immunohistochemistry.

To determine any potential transformation of the cells, pancreas organoids derived from Bl6 mice and cultured in our defined medium for at least 2 months were also directly transplanted into

the kidney capsule of nude, SCID or NSG mice. The grafts were harvested 2 weeks and 3 months later and were processed for paraffin section and H&E staining using standard techniques.

### β-galactosidase (LacZ) staining, immunohistochemistry and immunoflorescence

Tissues were fixed for 2 h in ice-cold fixative (1% Formaldehyde; 0.2% Glutaraldehyde; 0.02% NP-40 in PBS0) and incubated O/N at RT with 1–2 mg/ml of X-gal (bromo-chloro-indolyl-galactopyranoside) solution as we described in Barker *et al* (2010). The stained tissues were transferred to tissue cassettes and paraffin blocks were prepared using standard methods. Tissue sections (4 μM) were prepared and counterstained with neutral red. For immunohistochemistry, tissues and organoids were fixed using formalin 4%, and stained using standard histology techniques as described (Barker *et al*, 2010). The antibodies and dilutions used are listed in Supplementary Table SI. Stained tissues were counterstained with Mayer's Hematoxylin. Pictures were taken with a Nikon E600 camera and a Leica DFDC500 microscope (Leica). For whole-mount immunofluorescence staining, organoids were processed as described in Barker *et al* (2010).

Tissue sections (4 μM) or cryosections from kidney capsule grafts were processed for immunofluorescent staining using standard procedures. For the paraffin-embedded kidney capsule grafts citrate retrieval was performed. Antibodies and dilutions are listed in Supplementary Table SI. Nuclei were stained with Hoechst33342 (Molecular Probes).

### Microarray

For the expression analysis of pancreas cultures, total RNA was isolated from Sox9$^+$ duct cells (isolated as described in Supplementary Figure S5), acinar and islets cells (prepared from whole pancreas after collagenase dissociation), whole adult pancreas and pancreas organoids cultured in our defined medium, using Qiagen RNAase kit following the manufacturer's instructions. Five hundred nanograms of total RNA were labelled with the low RNA Input Linear Amp kit (Agilent Technologies, Palo Alto, CA). Universal mouse Reference RNA (Agilent) was differentially labelled and hybridized to the tissue or cultured samples. A 4 × 44K Agilent Whole Mouse Genome dual colour Microarray (G4122F) was used. Labelling, hybridization and washing were performed according to Agilent guidelines. Microarray signal and background information were retrieved using the Feature Extraction software (V.9.5.3, Agilent Technologies). The hierarchical clustering analysis was performed in duct, acinar, islet and organoid arrays after *in silico* subtraction of the pancreas gene array. A cutoff of two-fold differentially expressed was used for the clustering analysis. GSEA was performed according to Subramanian *et al* (2005). The gene lists and gene sets used for the analysis are all provided in Supplementary Datasets 1–6. GEO accession number is GSE50103.

### RT-PCR and qPCR analysis

RNA was extracted from cell cultures or freshly isolated tissue using the RNeasy Mini RNA Extraction Kit (Qiagen) or TRIzol (Invitrogen) respectively, and reverse transcribed using SuperScript II Reverse Transcriptase (Invitrogen). All targets were amplified (40 cycles) using gene-specific Taqman primers and probe sets (Applied Biosystems, London, UK). Data were analysed using the Sequence Detection Systems Software, Version 1.9.1 (Applied Biosystems). For *Neurog3*, cDNA was amplified in a thermal cycler (GeneAmp PCR System 9700; Applied Biosystems) as previously described (Huch *et al*, 2009). Primers used are listed in Supplementary Table SII.

### Image analysis

Images of cultivated cells were acquired using either a Leica DMIL microscope and a DFC420C camera or a Nikon TE2000 inverted automated fluorescence microscope with motorized table and controlled by the NIS elements AR software. Immunofluorescence images were acquired using an upright Zeiss Axioplan2 fluorescence microscope with Hamamatsu C10600 ORKA-R2 camera or a confocal microscope (Leica, SP5) or a confocal microscope (Leica, SP8) or a confocal multiphoton Zeiss LSM710 NLO with the TiSa laser microscope. Images were analysed using the Leica LAS AF Lite software (Leica SP5 confocal) or Smartcapture 3 (version 3.0.8). Confocal images were processed using Improvision VolocityLE and Zeiss Zen softwares.

### Data analysis

All values are represented as mean ± standard error of the mean (s.e.m.). Mann–Whitney non-parametric test was used. $P < 0.05$ was considered as statistically significant. In all cases, data from at least three independent experiments were used. All calculations were performed using the SPSS package.

### Supplementary data

Supplementary data are available at *The EMBO Journal* Online (http://www.embojournal.org).

## Acknowledgements

We thank Maaike van den Born, Stieneke van der Brink, Ann Demarre, Gunter Leuckx and Geert Stangé for technical assistance. The Hubrecht Imaging Center for imaging assistance. This work was supported by grants to MH (EU/236954) and SFB (EU/232814) and JHvE (Ti Pharma/T3-106), PB (EMBO fellowship, EFSD/JDRF grant), HH (EU-HEALTH-F5-2009-241883, Dutch Diabetes Foundation 2007.16.001, NFSR G000609N10, NIH 1U01DK089571-01, support from the Innovative Medicines Initiative Joint Undertaking under grant agreement n° 155005 (IMIDIA), resources of which are composed of financial contribution from the European Union's Seventh Framework Programme (FP7/2007–2013) and EFPIA companies in kind contribution). The Hubrecht Institute received financial support from the DON Foundation and Dutch Diabetes Research Foundation.

*Author contributions*: Experiments were conceived and designed by MH, PB, SFB, TS, HH, and HC. All experiments were performed as follows: MH, PB, TS: PDL and histology; MH, PB, SFB, TS: pancreatic duct cultures and *in vitro* characterization; MH: clonal culture of Lgr5 cells; microarray analysis; PB: prospective isolation and culture of purified clonal populations; morphogenetic *in vivo* assay; SFB: *in vitro* differentiation; microarray analysis; TS: development of the culture system; MH, PB, SBJ: direct transplantation. *Lgr5* sortings were performed by MvdW. AG and KH helped with PCR experiments and karyotypes. MS helped with lineage labelling mice and cell cultures. JM helped in the establishment of the differentiation protocol. CJML and FR performed kidney transplants. HB helped with the immunohistochemistry analysis of the kidney grafts. JHvE, EdK and RGJV discussed the project. MH, SFB, PB and TS analysed the data. JS and VSWL help with the microarray data. MH, SFB, PB, HH and HC wrote the manuscript, the other authors commented the manuscript.

## Conflict of interest

MH, TS and HC are inventors on a patent application related to this work.

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
