## [Review Process File · The EMBO Journal]

Manuscript EMBO-2013-84684

Unlimited in vitro expansion of adult bi-potent pancreas progenitors through the Lgr5/R-spondin axis

Meritxell Huch, Paola Bonfanti, Sylvia F Boj, Toshiro Sato, Cindy JM Loomans, Marc van de Wetering, Mozhddeh Sojoodi, Vivian SW Li, Jurian Schuijers, Ana Gracanin, Femke Ringnalda, Harry Begthel, Karien Hamer, Joyce Mulder, Johan H van Es, Eelco de Koning, Robert GJ Vries, Harry Heimberg and Hans Clevers

Corresponding author: Hans C. Clevers, Hubrecht Institute

Review timeline:

Submission date:	03 February 2013
Editorial Decision:	04 March 2013
Revision received:	15 August 2013
Accepted:	16 August 2013

Transaction Report:

Editor: Thomas Schwarz-Romond

1st Editorial Decision

04 March 2013

Thank you very much for submitting your study on the in-vitro expansion of adult pancreas precursors for consideration to The EMBO Journal editorial office.

The attached comments reflect the (not unexpected) division among the referees regarding conceptual novelty of the approach. Importantly, their comments emphasize the current lack of a robust model to maintain/expand adult pancreas progenitors that would yield great therapeutic potential.

Based on this strength, we would be delighted to evaluate a thoroughly revised/probably slightly refocussed study.

Given inherent discrepancies/difficulties in the area of differentiating glucose-responsive, insulin producing cells however, all referees demand substantial further experimentation to establish the differentiation potential (and thus therapeutic relevance) of the Lgr5+ population.

As their remarks are specific in what would be essential to substantiate both the in-vitro as well as the in-vivo features/capabilities, there is no reason to repeat these here in great detail.

Assuming that you might already have quite a very good handle on realistic/feasible amendments, I

suggest to discuss/agree upon crucial experiments upfront to avoid unnecessary delays for this study (preferably outlined in an E-mail, as to ensure transparency of the process and the possibility to double-check with some of the referees, if deemed necessary).

Please be reminded that The EMBO Journal considers only one round of revisions and the ultimate decision on publication will dependent on the outline and strength of your revised manuscript.

I am very much looking forward to your amended study and remain with best regards.

REFEREE REPORTS:

Referee #1:

In this manuscript the authors describe the isolation of an Lgr5+ cell type from the adult mouse pancreas. Wnt signalling and Lgr5 expression are not normally observed under normal physiological homeostasis however upon injury, caused by partial duct ligation, Wnt signalling is detected in the pancreas and a subset of pancreatic cells in pancreatic ducts upregulate Lgr5 expression. In vitro, the pancreatic ducts can be grown in previously reported Rspodin culture conditions. Isolated duct cells form budding cyst structures and can be passaged for more than 40-weeks. The resulting cells can be clonally passaged while maintaining an apparent capacity to differentiate into endocrine cells.

Overall, the manuscript is well organised and present interesting data especially concerning the activation of the WNT pathway upon duct ligation. This aspect is novel and provides important information. The possibility to grow the injury induced Lgr5+ cells is also interesting.

The characterisation of the organoid culture is conventional but it would have been useful to have a growth curve over 1-2 months to define the real proliferation potential of these cells. The karyotyping data should be reinforced by performing G-banding analyses. Counting chromosome is not enough. This aspect is really important for future applications of this culture system.

However the real weakness of this manuscript are the experiments describing the differentiation potential of the Pancreatic duct Lgr5+ cells. This part of the manuscript is questionable and need to be significantly reinforced to support the authors' conclusions.

The differentiation of Duct cells in vitro into endocrine cells is not convincing. PDX1 expression is not obvious and should be characterised by Q-PCR, Western blot and FACS analyses. Insulin expression is not induced (cells stained positive for Insulin by IF could be dead cells trapping insulin contained in the medium) while the expression of other endocrine markers is not analysed. What is happening with Glucagon and SST? Co-staining with these markers are necessary since functional ngn3 endocrine progenitors should be able to differentiate into all this lineage in vitro. Production of multi-hormonales cells could also be a major issue. In fact, the culture conditions used by the authors to drive differentiation of Pancreatic duct Lgr5+ cells are clearly not optimal and they must to screen a broader array of inducing factors. This is a major issue since it suggests that Pancreatic duct Lgr5+ cells have very limited if any capacity to differentiate into endocrine cells. Finally functional test are lacking. Insulin secretion upon glucose stimulation needs to be shown.

The in vivo experiments are similarly problematic. The co-culture system used is very complicated and have been used to drive differentiation of a diversity of cell type into pancreatic cells without any specificity. Figure 1D showing staining of INS in eGFP expression are not convincing. INS expression seems to be really weak in GFP cells. Could it be dying cells trapping Insulin? What about cell fusion? What is happening for Glucagon, SST and amylase? .

It would have been easier for the authors to transplant Pancreatic duct Lgr5+ cells in the kidney capsule without any other tissue and then wait for 10-16 weeks before analysing the differentiated cells. This protocol gives good results with PDX1 pancreatic progenitors generated from hESCs. What happens with their cells?

Referee #2:

This manuscript reports a method for long-term culture and expansion of pancreatic duct cells as 3D-organoid structures upon the activation of the Wnt-Lrp5-Rspo signaling axis. Similar culture conditions have been successfully used by the same group for long-term expansion of adult intestine, stomach and liver cells. In the adult pancreas, Wnt signaling is inactive and the stem cell marker *Lgr5* is not expressed under physiological conditions. *Lgr5* is upregulated upon pancreatic duct ligation, marking a potential pancreatic progenitor pool within the ductal tree. Importantly, clonal pancreas organoids, containing *Lgr5*⁺ cells, once established in culture can be induced to differentiate into duct and endocrine cell progenitors in vitro. The authors conclude that the *Lgr5*⁺ pancreatic bipotent progenitor cells represent a model system complementary to ESCs- and iPSCs-based strategies to cure diabetes.

Overall, this study is well conducted and provides a valuable model for studying pancreatic development *ex vivo*. It is indeed true that a robust long-term culture system capable of maintaining and expanding adult non-transformed pancreas progenitors is missing in the field. Moreover, previous attempts to establish pancreas cultures yielded to cultures of heterogeneous populations, which undergo senescence over time. However, I find that the characterization of the *Lgr5*⁺ pancreas organoid cultures, as presented here, is not complete and some specific points should be addressed.

Specific concerns with the manuscript are listed below:

1) The pancreas organoid cultures display some degree of heterogeneity. For instance, in clonal pancreatic organoids expanded from *Lgr5*-LacZ knock in mice most of the cells become *Lgr5*⁻ after few passages in culture (Fig. 2). The authors showed that most of pancreatic organoid cells are positive for *Pdx1* and *Sox9* pancreatic progenitor markers (Fig. 5A) and only a sub-fraction retain *Lgr5* expression (Fig. 2). Is the *Lgr5*⁺ population the only fraction capable of self-renewal? Do the *Lgr5*⁺ cells divide asymmetrically and give rise to the *Lgr5*⁻ cells that subsequently differentiate? If yes, can the authors trace them and determine what do they become?

2) To further characterize the pancreas organoid cultures, the authors performed a microarray analysis. On the basis of the hierarchical clustering analysis shown in Fig. 5 they conclude that clonal pancreas organoid cultures resemble adult pancreas. In particular, the cells display ductal cell-specific gene signatures but not endocrine and/or acinar identity. However, by looking at the heat map, many obvious differences are visible between adult pancreas and pancreas organoid cultures gene expression profiles. First, I recommend expanding the data analysis. For instance, it would be more informative to compare the gene expression profile of pancreas organoid cultures versus adult and also embryonic pancreases rather than to completely unrelated tissue such as muscle or fat. Also, the authors should provide some information or at least discuss the most obvious differences between the cells and adult pancreas (see upper part of the heat map above the ductal cluster). What are these genes?

3) The authors devised two strategies to differentiate the "ductal-like" pancreatic organoid cultures along the endocrine lineage either in vitro or in vivo. The in vitro strategy appears well thought, but the characterization of the differentiated endocrine progenitors is very preliminary. The analysis should be performed by quantitative RT-PCR (instead of semiquantitative RT-PCR - as shown in Fig. 6C -), expanded to more markers and include more extensive immunofluorescence analysis to appreciate i. the efficiency of differentiation and the ii. 3D-architecture of the differentiating organoids.

On the other hand, I have several concerns about the ability and efficiency of the organoid pancreatic cultures to differentiate in vivo. To date, ESC-, iPSC- or EP-derived pancreatic progenitor cells have been reported to differentiate into beta-cell like cells when transplanted alone into the kidney capsule. Do the *Lgr5*⁺ cells are capable to differentiate in vivo only when transplanted together with WT embryonic pancreatic cells? Have the authors tried the transplantation assay into kidney capsule without WT embryonic pancreatic cells? Have they tried to transplant the cells in vivo after exposing them to the in vitro differentiation protocol? Moreover, the cells should be transplanted in vivo into diabetic mouse models, in which hyperglycemia might help

to further push their differentiation into endocrine cells. These are all important experiments that need to be performed before making any conclusion about the *in vivo* potentials of the cells and any comparison to the available model systems based on ES or iPS cells.

4) According to standard criteria, stable cell line should be capable of indefinite growth and could be re-cultured after freezing. It is important to specify if the pancreatic organoid cultures can be frozen. Also, do the pancreatic organoid cells undergo any sign of transformation in culture? This should be checked in the *in vivo* transplantation assay.

Given the concerns delineated above, the manuscript cannot be accepted for publication in The EMBO Journal in its present form, however this reviewer would be happy to re-review a version that addresses the points raised above.

Referee #3:

This is an interesting paper that addresses the role of Lgr5+ cells in the expansion of pancreatic progenitor cells. The authors have used an approach similar to their own report in Nature (Huch et al. Nature 494:247-250, 2013). Thus, this series of studies is not novel but the same approach applied to another organ system!

I have a few concerns directly relevant to the studies:

1. A recent paper (Rankin MM, et al. Diabetes Jan 24, 2013) indicates that PDL (pancreatic duct ligation) does not enhance beta cell proliferation. This is in contradiction to the current studies. This needs to be explained convincingly by the authors.

2. The use of insulin in immunohistochemistry should be avoided because of the observation that insulin can be taken up passively and confound the data (Rajagopal et al. Science 299:5605, 2003). Thus, all experiments with insulin should be replaced with immunohistochemistry for C-peptide.

3. What is the functional status of the cells that have the so-called bi-potential capacity to generate exocrine and endocrine progenitors? Are they glucose responsive? This is critical if the authors want to support their claim that they are generating authentic mature endocrine progenitors. This becomes critical given that some studies cannot show that beta cells are produced in response to PDL (see comment #1).

1st Revision - authors' response

15 August 2013

Please find a revised version of our manuscript (EMBOJ-2013-84684R) entitled '*Unlimited in vitro expansion of adult bi-potent pancreas progenitors through the Lgr5/Response axis*'

Before addressing the individual comments of the reviewers, we would like to discuss the revised manuscript from a more general perspective: As a major change, we have put less emphasis on the *in vitro* differentiation data. Additionally, we explain in detail our novel morphogenetic assay and present the *in vivo* differentiation data as a proof of concept that an expanding, clonal adult duct population remains bi-potent.

As indicated in our title, the main message of this manuscript is not the production of endocrine cells from duct cells. The efficient generation of hormone-producing cells (insulin, glucagon or somatostatin) *in vitro* is a major challenge to the field, not likely accomplished in a single study, but will involve multiple technological developments. The current study describes one, essential step: the efficient clonal expansion of bi-potent progenitors from adult Lgr5 duct cells.

Let us summarize why we believe that the study is original and significant.

- 1) It describes a progenitor/stem cell population of Wnt-responsive, Lgr5-expressing cells that appears only in the regenerating pancreas. This is the first example of this type of “induced” progenitor cells in the pancreas.
- 2) It shows that *in vitro*, duct cells up-regulate Lgr5 and become activated. This Lgr5 population is the main population that self-renews in culture.
- 3) It describes a morphogenetic assay in which dissociated embryonic pancreas is mixed with the expanded duct cultures and directly transplanted (without *in vitro* incubation) into the mice. This assay has been adapted from the skin field and is novel to the pancreas field. It demonstrates (1) that a developing pancreas can be induced to differentiate *in vivo* into a mono-hormonal tissue and (2) that the embryonic pancreas contains all the necessary factors to induce adult duct cells to differentiate towards 3 endocrine lineages (somatostatin, glucagon and insulin).
- 4) It demonstrates that expanded duct cells are bi-potent. Clonal adult duct cells placed in an environment that contains all the signals for pancreas differentiation (embryonic pancreas) generate consistently duct and mono-hormonal cells (Insulin, glucagon or Somatostatin). This effect stimulated by external factors is also intrinsic to the duct cells, since when no signalling cues are present (direct transplantation into the kidney capsule) the cells can generate either duct cells (at high frequency) or endocrine (insulin and ChrA), albeit at low frequency.
- 5) It challenges the concept that pancreas duct cells are subject to the Hayflick limit. This clonal expansion opens up many experimental avenues, such as the establishment of disease models from patient cells *in vitro*, and the safe genetic (clonal) modification of pancreas stem cells for therapeutic purposes.

We would like to take this opportunity to thank you and the referees for the constructive suggestions that have resulted into an improved new manuscript.

We hope that –with these changes and additions- EMBO journal will now be able to publish our study.

Point by Point Response

Referee #1:

In this manuscript the authors describe the isolation of an Lgr5⁺ cell type from the adult mouse pancreas. Wnt signalling and Lgr5 expression are not normally observed under normal physiological homeostasis however upon injury, caused by partial duct ligation, Wnt signalling is detected in the pancreas and a subset of pancreatic cells in pancreatic ducts upregulate Lgr5 expression. *In vitro*, the pancreatic ducts can be grown in previously reported Rspodin culture conditions. Isolated duct cells form budding cyst structures and can be passaged for more than 40-weeks. The resulting cells can be clonally passaged while maintaining an apparent capacity to differentiate into endocrine cells.

Overall, the manuscript is well organised and present interesting data especially concerning the activation of the WNT pathway upon duct ligation. This aspect is novel and provides important information. The possibility to grow the injury induced Lgr5⁺ cells is also interesting. The characterisation of the organoid culture is conventional but it would have been useful to have a growth curve over 1-2 months to define the real proliferation potential of these cells.

A: We appreciate the reviewer's suggestion. The data in the original manuscript included a growth curve up to 32 days (~1 month) of cultures derived from single sorted cells. We have now extended the growth curve of the organoid cultures to 3 months. As expected, the cells maintain an exponential growth curve at all time points analysed: days 6-20 (P1-P3), day 37-51 (P5-P7) and day 78-100 (P10-12). We have also now calculated the doubling time of the cells at the different passages and we observe that it is essentially maintained in the range of 53-60 hours. We now provide this new growth curves and doubling times in the new Figure 2C and we have moved the original growth curve panel of single cells to Supplementary Figure 3D.

Furthermore, the cultures have been serially passaged in a 1:4-1:5 ratio every week for >9 months (from organoids Fig. 2B) or from single cells (Fig. 3D and Fig. 4D), thus indicating that the cells maintain an exponential growth also beyond the 3 months period of the growth curve analysis.

The karyotyping data should be reinforced by performing G-banding analyses. Counting chromosome is not enough. This aspect is really important for future applications of this culture system.

A: Unfortunately, our genetics department has no experience with G-banding of mouse chromosomes (that are all acrocentric and of similar size and therefore exceedingly difficult to identify individually). For this reason, chromosome counts are widely used as the golden standard for (for instance) genotyping ES cells in the generation of knock-out mice. We believe that to fully assess any possible transformation or genetic changes due to long term culturing, the gold-standard analysis would be to perform whole genome sequencing in clonal cultures serially taken from different time points expanded for months in culture. However, this experiment is currently ongoing and it will be presented in a new manuscript.

As to potential transformation of the cells in culture, we have transplanted pancreas organoids (>2 months old) into the kidney capsule of 3 different immunocompromised mouse strains (NSG, nude and Nod-SCID, n=9). We did not observe any signs of malignant transformation since only normal appearing ductal-structures are detected 3 months after engraftment (new Supplementary Figure 2C-D).

Also, as demonstrated in the previous version of the manuscript, we did not find any signs of aneuploidy in the chromosomal counts performed at different time points (original figure3). Furthermore, the observation that the withdrawal of growth stimuli results in proliferation arrest (original figure 2) and re-starts the differentiation program is another indication that the cultures do not undergo transformation after long term expansion.

We have now summarized these data (transplantation, chromosomal counts and growth stimuli withdrawal) all together in the new Supplementary Figure 2.

However the real weakness of this manuscript are the experiments describing the differentiation potential of the Pancreatic duct Lrg5+ cells. This part of the manuscript is questionable and need to be significantly reinforced to support the authors' conclusions.

The differentiation of Duct cells in vitro into endocrine cells is not convincing. PDX1 expression is not obvious and should be characterised by Q-PCR, Western blot and FACS analyses.

Insulin expression is not induced (cells stained positive for Insulin by IF could be dead cells trapping insulin contained in the medium) while the expression of other endocrine markers is not analysed.

What is happening with Glucagon and SST? Co-staining with these markers are necessary since functional ngn3 endocrine progenitors should be able to differentiate into all this lineage in vitro.

Production of multi-hormonal cells could also be a major issue. In fact, the culture conditions used by the authors to drive differentiation of Pancreatic duct Lrg5+ cells are clearly not optimal and they must to screen a broader array of inducing factors. This is a major issue since it suggests that Pancreatic duct Lrg5+ cells have very limited if any capacity to differentiate into endocrine cells.

Finally functional test are lacking. Insulin secretion upon glucose stimulation needs to be shown.

A: We would like to emphasize that the message of this manuscript is not the production of endocrine cells from duct cells in vitro or in vivo (see cover letter). The efficient generation of hormone-producing cells (insulin, glucagon or SST) in vitro appears to be the major challenge to the field, not likely accomplished in a single study. This will likely involve multiple technological steps. The current study describes one, essential step: the efficient clonal expansion of bipotent progenitors from adult Lgr5 duct cells.

As we do want to make the explicit statement that clonal offspring of Lgr5+ duct cells is bipotent and can be 'primed' toward an endocrine fate in vitro, we present the expression of the early endocrine markers Pdx1, Neurog3 and ChrA in culture to document the 'endocrine priming' of the

cells prior to direct transplantation (Supplementary Fig 9B). We present new, improved data for the *in vivo* differentiation in the new Figures 6 and 7 and Supplementary fig 9C-F.

As we are not producing functional beta-cells in vitro, we are also not evaluating insulin secretion.

The *in vivo* experiments are similarly problematic. The co-culture system used is very complicated and have been used to drive differentiation of a diversity of cell type into pancreatic cells without any specificity.

A: The morphogenetic pancreas assay used in this manuscript is developed by us and is novel. It does not involve any co-culture in vitro. We are not aware (and the reviewer does not quote any reference) about the fact that any cell type (except embryonic pancreatic progenitors) under similar conditions will turn into beta cells.

This assay is based on removing the embryonic pancreata at E13.5 (mouse) or E15 (rat) that are dissociated and mixed at a fixed ratio with cultivated eGFP+ or CFP+ organoids dissociated to single cells. The aggregate is then directly transplanted (without co-culturing) under the kidney capsule of nude mice for a period of at least 2 weeks where the cultivated cells develop towards mature functional structures (CK+ cells and Gcg+, Sst+, INS+ and Cppt+ single hormonal cells).

A similar approach has been successfully used, and accepted as an in vivo functional assay for morphogenesis of other tissues (skin and thymus; Bonfanti et al, Nature 2010). In stark contrast to co-culture of pancreatic explants with various cell types, this assay allows full development of all pancreatic lineages from the embryonic stage with the advantage to allow development and maturation for much longer time (weeks) in vivo. The resulting structure is a pancreas organ that has developed ectopically with the advantage of the ability to incorporate cultivated cells that can actively interact with the microenvironment where they receive the proper signals for differentiation. In fact, the system provides all the signals that drive early progenitors to differentiate towards mature cells. We have developed this assay with the intent to disclose the full fate potency of cultivated adult duct cells after extensive expansion in vitro.

This is now explained in more detail in the revised manuscript. Also, we now provide images showing that the embryonic pancreas transplanted into the kidney capsule results in a structure that contains all the cell types present in a pancreas organ, including ducts, islets and acinar cells (Supplementary Figure 7A).

Figure 1D showing staining of INS in eGFP expression are not convincing. INS expression seems to be really weak in GFP cells. Could it be dying cells trapping Insulin?

What is happening for Glucagon, SST and amylase?

A: We now provide images (that should be convincing), further illustrated by the higher resolution that we have added (Figure 6C-E and Supplementary Figure 7C-E). We had chosen to show these panels at a low magnification just to illustrate that WT insulin cells have a similar distribution and morphology in the same area of the ectopic pancreas. Moreover, cell and nucleus morphologies do not indicate these are dying cells (no nuclear pycnosis or fragmentation), and both INS and SYP staining have the typical granular pattern of secretory cells. In addition, we now provide IHC images of Ins+ cells also positive for Cppt, and not only for Syp (Figure 6D,E and Supplementary Fig 7E).

Regarding insulin trapping: In this type of assay the follow up time window (2-3 weeks after transplantation) is long enough to not allow the detection of cells that in vitro could have been trapping insulin. We have reproduced the same results using a new reporter mouse that express CFP under the control of E-Cadherin promoter and thus have a membrane localization of the reporter (Figure 6D,E and Supplementary Figure 8A,B). These images unequivocally identify pancreas culture-derived CFP+ cells that have cytoplasmic expression of both INS and mouse-specific Cppt even when engrafted in a rat pancreas microenvironment where endogenous INS+ cells are negative for the mouse-specific anti-Cppt antibody (Figure 6D,E and Supplementary Figure 8B,C).

We now provide images that document Glucagon+ and Sst+ single positive cells in the in vivo assay in addition to Ins+Cppt+ cells (Figure 7). No Amylase+eGFP+ cells have been detected in the ectopic pancreas, the reason why we define the adult cultivated cells as bipotent.

What about cell fusion?

A: In order to exclude the possibility of fusion between cultivated cells and WT endocrine cells, we have performed the same morphogenetic assay using embryonic pancreas from rat instead of from mouse. We now provide images of differentiated beta cells that are positive for mouse Cppt, while the rat beta cells are not labelled, demonstrating absence of fusion with rat beta cells (Figure 6D). The specificity of this antibody both in the ectopic pancreas and in the adult rat pancreas is shown in Supplementary Fig. 8B,C.

It would have been easier for the authors to transplant Pancreatic duct Lgr5+ cells in the kidney capsule without any other tissue and then wait for 10-16 weeks before analysing the differentiated cells. This protocol gives good results with PDX1 pancreatic progenitors generated from hESCs. What happens with their cells?

A: We have now transplanted 2-months old pancreas organoids from both, B16 wt mice or EcadCFP mice directly into the kidney capsule of immune-deficient mice. Prior to transplantation the cultures were 'primed' in vitro to express some early progenitor markers (Neurogn3 and ChrA), by culturing them in the reported Kroon medium (Kroon et al, 2008) supplemented with ILV and DBZ as described in Material and Methods. One and two months after transplantation Krt19 positive cells were readily detect all over the graft. Also, at much lower efficiency, Insulin+/Cpeptide+ cells, as well some ChrA+ cells are detected. This new data reinforce our conclusion that indeed, expanded duct cells can generate not only duct but also endocrine cells. These additional data are presented in Supplementary Fig 9.

We want to stress that this direct in vivo differentiation has a lower efficiency than when the cells are mixed with the embryonic pancreas, thus indicating that only when the signalling is optimal the cells develop their full differentiation potential. Identifying this crucial signalling in the embryonic pancreas niche will be a crucial next step to develop new strategies to better differentiate these cells in vitro and accomplish their therapeutic potential. Importantly, the differentiation efficiency reported by most published protocols of huESC towards endocrine cells is very low (<1). It is important to keep in mind that the amount of starting cells is dramatically higher in the case of huES than in our experiments with the adult ductal progenitors, which makes the two systems not directly comparable. We have changed the text accordingly and removed any comparison between ESC and our adult duct bi-potent progenitors.

Referee #2:

This manuscript reports a method for long-term culture and expansion of pancreatic duct cells as 3D-organoid structures upon the activation of the Wnt-Lrp5-Rspo signalling axis. Similar culture conditions have been successfully used by the same group for long-term expansion of adult intestine, stomach and liver cells. In the adult pancreas, Wnt signalling is inactive and the stem cell marker Lgr5 is not expressed under physiological conditions. Lgr5 is upregulated upon pancreatic duct ligation, marking a potential pancreatic progenitor pool within the ductal tree. Importantly, clonal pancreas organoids, containing Lgr5+ cells, once established in culture can be induced to differentiate into duct and endocrine cell progenitors in vitro. The authors conclude that the Lgr5+ pancreatic bipotent progenitor cells represent a model system complementary to ESCs- and iPSCs-based strategies to cure diabetes.

Overall, this study is well conducted and provides a valuable model for studying pancreatic development ex vivo. It is indeed true that a robust long-term culture system capable of maintaining

and expanding adult non-transformed pancreas progenitors is missing in the field. Moreover, previous attempts to establish pancreas cultures yielded to cultures of heterogeneous populations, which undergo senescence over time. However, I find that the characterization of the Lgr5+ pancreas organoid cultures, as presented here, is not complete and some specific points should be addressed.

Specific concerns with the manuscript are listed below:

1) The pancreas organoid cultures display some degree of heterogeneity. For instance, in clonal pancreatic organoids expanded from Lgr5-LacZ knock in mice most of the cells become Lgr5- after few passages in culture (Fig. 2). The authors showed that most of pancreatic organoid cells are positive for Pdx1 and Sox9 pancreatic progenitor markers (Fig. 5A) and only a sub-fraction retain Lgr5 expression (Fig. 2).

A: In figure 2 the cultures were actually not clonal, but derived from duct fragments from non-damaged pancreas of Lgr5-LacZ mice or Axin2 LacZ mice.

We agree with the reviewer that in our culture, each organoid consists of a mix of cell types, even when derived from a single Lgr5+ cell (Figure 4E). Between organoids, this picture is very homogeneous. We have described the same phenomenon in intestinal organoids, where stem cells make up a minority of the cells, while we observe all epithelial cell types at roughly normal frequencies. This fact is mirrored in the Lgr5 staining of pancreas organoids, where not all cells within an organoid are Lgr5+ (Figure 2D and 4E). This mimics what happens in vivo, upon PDL damage, where almost all the duct cells are Wnt-responsive (Axin2+) but only a subset of them are Lgr5+ (Figure 1D vs 1F). Similarly, the proliferating capacity is restricted to a subset of cells within the organoid (Figure 5A).

Is the Lgr5+ population the only fraction capable of self-renewal?

A: In original figure 5, we had assessed the clonogenic capacity of the Lgr5^{+ve} population and demonstrated that these Lgr5^{+ve} cells have higher colony formation efficiency than their Lgr5^{-neg} counterparts (16.03% vs 1.6%).

We have now generated new Lgr5^{+ve} and Lgr5^{-neg} single cell derived clones and assessed their long term self renewal potential and Lgr5 expression. The results, presented in the new Supplementary Figure 6, demonstrate that clonal Lgr5-negative populations once they start growing into organoids in culture, will re-express Lgr5 and, subsequently, behave as an Lgr5^{+ve} clone, that will self-renew in vitro long term (at least for >2 months in culture). This Lgr5 re-expression resembles what happens when the culture is started from a healthy pancreas (non-PDL, either duct fragments or Sox9 ductal cells). In the healthy freshly isolated tissue, Lgr5 is not expressed but, upon culturing, the cells start expressing Lgr5, mimicking what happens in vivo, upon damage (PDL).

These results resemble our previous observations in the intestinal epithelium where committed progenitor cells that do not expressed Lgr5 (e.g. Dll1 cell), upon damage of the intestinal epithelium, or upon in vitro culture, can convert back into a Lgr5+ cells at low frequency and then behave as true stem cells in the gut (van Es et al. Nat. Cell Biology 2012).

Do the Lgr5+ cells divide asymmetrically and give rise to the Lgr5- cells that subsequently differentiate?

If yes, can the authors trace them and determine what do they become?

A: We agree with the reviewer that this is indeed a very interesting question, but this would be a study in itself, beyond the scope of this paper. We have performed similar studies for the gut Lgr5 cells and find that they do not divide asymmetrically but rather that their fate is determined by the location of the daughter cells, independent of cell division (Snippert et al, Cell 2011). These studies are -however- not simple.

2) To further characterize the pancreas organoid cultures, the authors performed a microarray analysis. On the basis of the hierarchical clustering analysis shown in Fig. 5 they conclude that clonal pancreas organoid cultures resemble adult pancreas. In particular, the cells display ductal cell-specific gene signatures but not endocrine and/or acinar identity. However, by looking at the heat map, many obvious differences are visible between adult pancreas and pancreas organoid cultures gene expression profiles. First, I recommend expanding the data analysis. For instance, it would be more informative to compare the gene expression profile of pancreas organoid cultures versus adult and also embryonic pancreases rather than to completely unrelated tissue such as muscle or fat.

A: We agree with the reviewer. We have now generated new microarray data comparing the profile of the pancreas organoids to the 3 main populations of the pancreas (ductal cells, islets and acinar cells).

The overall gene expression profile of the organoid cultures clusters with the duct cell arrays, whereas it does not cluster with the gene profiles of acinar or endocrine cells. Of note, among the genes whose expression pattern do cluster between the duct pancreatic cells and the organoids, we find the well-known duct markers Sox9, Krt7, Krt19 and Spp1. This new cluster analysis is now presented in Figure 5B and supplementary dataset S1.

Also, we have now performed Gene Set Enrichment analysis (GSEA) between the genes enriched in the organoids and adult ductal and embryonic pancreas populations. We confirm now that the organoid cultures are enriched in genes specifically expressed in adult Sox9+ pancreatic duct cells while there is no enrichment in genes expressed in acinar cells or in genes expressed during the developing pancreas at E14.5 or E17.5 (Figure 5D-E). Importantly, we also observe enrichment in genes previously reported in small intestinal and pancreas stem cells, i.e. Lgr5, Prom1, Sox9 and Lrig1. Overall, these results confirm the adult pancreas duct nature of our pancreas progenitor cultures.

Since we also include the differential expression profile of the organoids compared to the pancreas tissue (see below), we have removed the old comparison to the non-related organs like muscle or fat, as suggested by the reviewer, and concentrated only on the profiles of pancreas related cells types.

The list of genes in common between these gene sets and the pancreas organoids are listed in the supplementary datasets S3-S6.

Also, the authors should provide some information or at least discuss the most obvious differences between the cells and adult pancreas (see upper part of the heat map above the ductal cluster). What are these genes?

A: We had previously included a direct comparison of the gene expression of the pancreas organoids and adult pancreas (original Figure 5D). We now include a detailed list of genes differentially expressed (at least 2fold) between pancreas organoids and pancreas tissue in the supplementary dataset S2. Overall, the comparison of the gene expression profile of the pancreas organoids and the pancreatic confirms the segregation of the non-ductal pancreatic markers (like Sst, Ins2, Gcg and Amy) and the ductal markers (like Krt7, Tcf2 and Sox9) as shown in Figure 5C and Supplementary Dataset S2. Of note, the Wnt target genes Lgr5, Cnd1 and Axin2 are also specifically highly expressed in the organoids (Figure 5C, Supplementary Dataset S2).

3) The authors devised two strategies to differentiate the "ductal-like" pancreatic organoid cultures along the endocrine lineage either in vitro or in vivo.

The in vitro strategy appears well thought, but the characterization of the differentiated endocrine progenitors is very preliminary. The analysis should be performed by quantitative RT-PCR (instead of semiquantitative RT-PCR - as shown in Fig. 6C -), expanded to more markers and include more extensive immunofluorescence analysis to appreciate i. the efficiency of differentiation and the ii. 3D-architecture of the differentiating organoids.

A: We emphasize that we do not aim to generate hormone-producing cells (insulin, glucagon or SSt) in vitro. This is a very ambitious aim clearly out of the scope of this paper. The previous in vitro data was presented to demonstrate the potential of the system to generate endocrine precursors. See response to Rev#1. Therefore, we have now restricted the in vitro differentiation results to the

expression pattern of the cells that are going to be further directly transplanted in vivo. See response below.

On the other hand, I have several concerns about the ability and efficiency of the organoid pancreatic cultures to differentiate in vivo. To date, ESC-, iPSC- or EP-derived pancreatic progenitor cells have been reported to differentiate into beta-cell like cells when transplanted alone into the kidney capsule. Do the Lgr5⁺ cells are capable to differentiate in vivo only when transplanted together with WT embryonic pancreatic cells? Have the authors tried the transplantation assay into kidney capsule without WT embryonic pancreatic cells? Have they tried to transplant the cells in vivo after exposing them to the in vitro differentiation protocol? Moreover, the cells should be transplanted in vivo into diabetic mouse models, in which hyperglycemia might help to further push their differentiation into endocrine cells. These are all important experiments that need to be performed before making any conclusion about the in vivo potentials of the cells and any comparison to the available model systems based on ES or iPS cells.

A: We thank the reviewer for this suggestion. We have now transplanted 2-months old pancreas organoids from both, B16 wt mice or EcadCFP mice directly into the kidney capsule of immune-deficient mice. Prior to transplantation the cultures were primed in vitro to express some early progenitor markers (Neurogn3 and ChrA), by culturing them in the reported Kroon medium supplemented with DBZ and ILV. One and Two months after transplantation Krt19 positive cells were readily detect all over the graft. Also, at much lower efficiency, Insulin/Cpeptide + cells as well some ChrA⁺ cells are detected. These new data reinforce our conclusion that indeed, expanded duct cells can generate not only duct but also endocrine cells. These additional data are presented in Supplementary Fig 9. See also our answer to reviewer 1 on this point.

4) According to standard criteria, stable cell line should be capable of indefinite growth and could be re-cultured after freezing. It important to specify if the pancreatic organoid cultures can be frozen.

A: Interesting point. So far, all our organoid cultures (small intestine, stomach and liver) can be frozen and recovered after thawing while maintaining all their expansion and differentiation properties. We now provide data (Figure 1, rebuttal letter) of pancreas organoid cultures that have been frozen and stored in liquid nitrogen, and then thawed and expanded at least for 5 weeks by sub culturing them in a 1:4-1:5 ratio every week. We state this now in the manuscript "These culture conditions allowed the recovery of the cells after freeze-thawing" and present the data as Figure 1 at the end of this rebuttal letter, to be assessed by the reviewer.

Also, do the pancreatic organoid cells undergo any sign of transformation in culture? This should be checked in the in vivo transplantation assay.

A: Fair point, thanks. We believe that to fully assess any possible transformation or genetic changes due to long term culturing, the gold-standard analysis would be to perform whole genome sequencing in clonal cultures serially expanded for months in culture. However, this experiment is beyond the scope of this manuscript.

To give an answer to the reviewer's comment, we have followed the reviewer's suggestion and transplanted pancreas organoids (>2 months old) into the kidney capsule of 3 different immunocompromised mouse strains (NSG, nude and Nod-SCID, n=9). We did not observe any signs of malignant transformation since only normal appearing ductal-structures are detected 3 months after engraftment (new supplementary figure 2C-D).

Also, as demonstrated in the previous version of the manuscript, we did not find any signs of aneuploidy in the chromosomal counts performed at different time points (original figure3). Furthermore, the observation that the withdrawal of growth stimuli results in proliferation arrest (original figure 2) and re-starts the differentiation program is another indication that the cultures do not undergo transformation after long-term expansion.

We have now summarized these data (transplantation, chromosomal counts and growth stimuli withdrawal) in the new supplementary figure 2.

Given the concerns delineated above, the manuscript cannot be accepted for publication in The EMBO Journal in its present form, however this reviewer would be happy to re-review a version that addresses the points raised above.

Referee #3:

This is an interesting paper that addresses the role of Lgr5+ cells in the expansion of pancreatic progenitor cells. The authors have used an approach similar to their own report in Nature (Huch et al. Nature 494:247-250, 2013). Thus, this series of studies is not novel but the same approach applied to another organ system!

I have a few concerns directly relevant to the studies:

1. A recent paper (Rankin MM, et al. Diabetes Jan 24, 2013) indicates that PDL (pancreatic duct ligation) does not enhance beta cell proliferation. This is in contradiction to the current studies. This needs to be explained convincingly by the authors.

We used the PDL model only to damage the pancreas (a fully accepted procedure) to show that the proliferating duct cells are Wnt-responsive and that a subpopulation of these expresses Lgr5. We are aware about the controversy of the PDL model to induce beta cell proliferation, but we make no use of this phenomenon. A discussion of the beta cell controversy appears therefore outside the scope of the current paper.

2. The use of insulin in immunohistochemistry should be avoided because of the observation that insulin can be taken up passively and confound the data (Rajagopal et al. Science 299:5605,2003). Thus, all experiments with insulin should be replaced with immunohistochemistry for C-peptide.

A: Well taken. We now have added IHC for mouse specific Cppt that co-localizes with INS+ cells, so the possibility of passive pick up of cells is ruled out (Figure 6D,E and Supplementary Figure 7E). Also, as mentioned before (see response to reviewer #1), in this type of assay the follow up time window (2-3 weeks after transplantation) is longer enough to not allow detecting cells that in vitro could have been trapping any insulin. Furthermore, the co-localization of INS with mouse specific Cppt in 1 month old grafts after direct transplantation of the organoids reinforces that same notion (see response to reviewers 1 and 2).

More important, we have reproduced the same results using a new reporter mouse that express CFP under the control of E-Cadherin promoter (supplementary fig 8A) and thus have a membrane localization of the reporter (Figure 6D,E and Supplementary Fig. 7E). These images unequivocally identify CFP+ cells that have cytoplasmic expression of both INS and mouse specific Cppt engrafted in a rat pancreas microenvironment. The antibody used to detect Cppt is mouse specific (see supplementary figure 8B,C) indicating that the c-peptide detected in the grafts, where the organoid cells were mixed with rat embryonic pancreas, is indeed derived from the mouse transplanted cells and not from rat pancreas where endogenous INS+ cells are negative for the mouse/specific anti-Cppt antibody (Figure 6D).

Also, we now provide images that document Glucagon+ and Sst+ single positive cells in the in vivo assay in addition to Ins+Cppt+ cells (Figure 7). No Amylase+eGFP+ cells have been detected in the ectopic pancreas, the reason why we define the adult cultivated cells as bi-potent.

3. What is the functional status of the cells that have the so-called bi-potential capacity to generate exocrine and endocrine progenitors? Are they glucose responsive? This is critical if the authors want to support their claim that they are generating authentic mature endocrine progenitors. This becomes

critical given that some studies cannot show that beta cells are produced in response to PDL (see comment #1).

A: As these cells are bipotent progenitors they are -by definition- not differentiated and should not display functional glucose responsiveness or any of the other qualities of their fully differentiated offspring.

In our manuscript we demonstrate that ductal cells that are Wnt responsive and express Lgr5 proliferate, generating new duct and new Lgr5 positive cells. These progenitors can generate at least duct cells and endocrine cells, when the differentiation signals are appropriate. Also, they differentiate into INS⁺ Cppt⁺ cells, in 1 month-old grafts, after direct transplantation of the organoids (see response to reviewers 1 and 2) (new supplementary figure 9). Overall, these results demonstrate that the expanded adult progenitor cells do indeed have the potency to differentiate into both duct and endocrine lineages. Therefore, they are true bi-potent progenitors.

To our knowledge, no evidence has been reported for long-term expansion of adult pancreatic progenitors and the in vitro and in vivo assays utilized are sufficient, in our opinion, to show that they maintain differentiation potency.

Reviewer #2:

Figure 1

Figure 1 Organoids after freeze-thawing

Duct fragments from a pancreas of a wild-type mouse or single cell derived clones from an Lgr5-LacZ culture were seeded in matrigel and cultured as described in Material and Methods. Cultures that had been split at least 1 time (P1) were frozen using Recovery Cell culture Freezing medium (gibco) and standard protocol procedures and stored in liquid N₂ for 1 week. Then, the cultures were thawed using standard thawing methods and embedded in matrigel. Once the matrigel was solidified, complete medium was added and cells were allowed to grow. Cultures were serially passaged at a 1:4-1:5 ratio for the following weeks (1 split/week). Representative pictures of a clonal pancreas culture, frozen at P1, just after thawing (left), 1 day after after (middle) and 5 passages later (right).

2nd Editorial Decision

16 August 2013

Thank you very much for submission of your extensively revised and modified paper. I critically assessed the amendments and responses to the concerns raised from the original referees. I further discussed these with members of our editorial team.

Based on this, I am pleased to inform you about formal acceptance of your study for publication in The EMBO Journal.